# Molecular assays to detect the presence and viability of *Phytophthora ramorum* and *Grosmannia clavigera*

**Barbara Wong[1,2☯], Isabel Leal[3☯], Nicolas Feau[2], Angela Dale[2,4], Adnan Uzunovic[4], Richard C. Hamelin**[1,2]*

**1** Faculté de foresterie et géomatique, Institut de Biologie Intégrative et des Systèmes (IBIS), Université Laval, Québec, QC, Canada, **2** Department of Forest and Conservation Sciences, University of British Columbia, Vancouver, BC, Canada, **3** Pacific Forestry Centre, Natural Resources Canada, Victoria, BC, Canada, **4** FPInnovations, Vancouver, BC, Canada

☯ These authors contributed equally to this work.
* richard.hamelin@ubc.ca

**Data Availability Statement:** Data are available via Dryad: (https://doi.org/10.5061/dryad.jq2bvq861).

**Funding:** This work was supported by: Grant from Genome Canada (10106) to RCH; Grant from the

## Abstract

Wood and wood products can harbor microorganisms that can raise phytosanitary concerns in countries importing or exporting these products. To evaluate the efficacy of wood treatment on the survival of microorganisms of phytosanitary concern the method of choice is to grow microbes in petri dishes for subsequent identification. However, some plant pathogens are difficult or impossible to grow in axenic cultures. A molecular methodology capable of detecting living fungi and fungus-like organisms *in situ* can provide a solution. RNA represents the transcription of genes and can become rapidly unstable after cell death, providing a proxy measure of viability. We designed and used RNA-based molecular diagnostic assays targeting genes essential to vital processes and assessed their presence in wood colonized by fungi and oomycetes through reverse transcription and real-time polymerase chain reaction (PCR). A stability analysis was conducted by comparing the ratio of mRNA to gDNA over time following heat treatment of mycelial cultures of the Oomycete *Phytophthora ramorum* and the fungus *Grosmannia clavigera*. The real-time PCR results indicated that the DNA remained stable over a period of 10 days post treatment in heat-treated samples, whereas mRNA could not be detected after 24 hours for *P. ramorum* or 96 hours for *G. clavigera*. Therefore, this method provides a reliable way to evaluate the viability of these pathogens and offers a potential way to assess the effectiveness of existing and emerging wood treatments. This can have important phytosanitary impacts on assessing both timber and non-timber forest products of commercial value in international wood trade.

## Introduction

Wood and wood products can harbor microorganisms that can raise phytosanitary concerns in countries importing or exporting these products [1–3]. Various treatments have been developed and can be applied to eliminate organisms present in wood [3]. The two most widely

Genomic Research and Development (GRDI) initiative to IL; Natural Sciences and Engineering Research Council of Canada (NSERC) TRIA-Net (NETGP- 434810-12).

**Competing interests:** The authors have declared that no competing interests exist.

used methods are heat treatment and fumigation of timber and wood products performed according to international phytosanitary standards ISPM 15 and ISPM 39 [4]. These methods are efficient for the elimination of insects but their efficacy in eliminating microorganisms can vary and is not always well documented. To evaluate the efficacy of these treatments the presence of microorganisms must be assessed following treatment of the wood products, generally using approaches based on the isolation and cultures of the microorganisms. However, culture-based methods have limitations as it is now estimated that only a small fraction of the microorganisms present in natural environments can be grown on artificial media [5,6]. In addition, some fungi grow slowly and rely on complex nutrient requirements. This is the case of certain fungi that are outcompeted by fast-growing saprophytic species [7]. These short-comings could generate false negatives, i.e. the failure to detect microorganisms that are still viable following a treatment that is inefficient. An additional challenge is that the paucity and sometimes inadequacy of distinguishing morphological traits complicates the identification of microorganisms [8].

The use of molecular methods, in particular amplification of conserved genes from genomic DNA (gDNA), followed by amplicon sequencing and DNA barcoding (matching the unknown sequence by homology in public sequence databases to provide identity), have become the standard in identification of fungi and oomycetes [9,10]. Quantitative real-time PCR (qPCR), a method that uses fluorescent dyes for detection of target DNA molecules/nucleic acids during the amplification process has increasingly replaced conventional PCR [11,12]. However, molecular detection methods are generally based on the detection of pathogen gDNA and therefore aim at detecting the presence, but not the viability, of the organism. Since DNA is stable and does not rapidly degrade following cell death, these assays are not useful to assess the viability of the targeted organisms and thus cannot help in determining the efficacy of a phytosanitary treatment. In contrast, messenger RNA (mRNA) degrades more rapidly after cell death[13–15], and is only produced by metabolically active cells, making it suitable to specifically detect living microorganisms [16].

To develop qPCR assays that can detect and quantify pathogen mRNA as an indicator of viability, it is necessary to produce complementary DNA (cDNA), which is the double-stranded DNA synthesized from a single stranded RNA (e.g., mRNA template in a reaction catalyzed by the reverse transcriptase enzyme). Complementary DNA amplified by PCR can then be used to measure the expression of mRNA and serve as a marker of cell viability in eukaryotes. Because introns are spliced out in mature mRNA, transcripts can be distinguished from gDNA. It is therefore possible to design assays utilizing a probe or primer that spans the exon-exon junction so that it can anneal to mRNA but not gDNA. This allows detection and quantification of mRNA without cross-amplification of the gDNA [17,18].

Herein, we report the design and validation of assays that amplify and provide a relative quantification of the mRNA over gDNA of two eukaryotic microorganisms that can colonize wood cambium: the oomycete *Phytophthora ramorum*, causal agent of sudden oak death and sudden larch death [19,20], and the blue stain ascomycete fungus *Grosmannia clavigera*, an important symbiont of the mountain pine beetle [21,22]. Since these two organisms belong to different kingdoms we are using them as a case-study for the development of a method that will be used to assess the viability of infectious microorganisms following wood treatments and evaluate the efficacy of such treatments.

## Materials and methods

### Design of assays

The program "PHYLORPH" (PHYLogenetic markers for ORPHans) was used to reconstruct the gene alignments with intron/exons junctions for *P. ramorum* (Pr-102 [ATCC MYA-2949];

[23]) and other closely related species [24]. Two hundred and twenty eight conserved proteins for *P. ramorum* were identified by performing a BLAST (Basic Local Alignment Search Tool) search with the CEGMA (Core Eukaryotic Genes Mapping Approach) protein database [25] against the genome sequences of *P. lateralis* (GCA_000500205.2), *P. hibernalis*, *P. foliorum*, *P. syringuae* and *P. brassicae* [26] (Brett Tyler, Oregon State University, Personal communication). Based on their putative function related to basic metabolic processes and their high expression levels in transcriptome analyses [27–30], these genes are expected to be expressed under various conditions in living organisms, an important criterion in developing assays to assess viability. A total of 43 candidate gene alignments were obtained from which seven primer pairs targeting *P. ramorum* were designed, on two adjacent exons separated by a short intron sequence (<95 bp in length.), using Geneious (v8.1.6). Primer pairs were screened for specificity using PCR with electrophoresis on gDNA extracted from cultures of 10 *Phytophthora* species from clade 8 (*P. ramorum*, *P. lateralis*, *P. brassicae*, *P. cryptogea*, *P. drechsleri*, *P. foliorum*, *P. hibernalis*, *P. porri*, *P. primulae* and *P. syringae*) which are all closely related [31].

The same procedure was performed on the *G. clavigera* genome (isolate kw1407;[27]) using the FUNYBASE protein database [32] against the genomes of *Leptographium longiclavatum*, *Neurospora crassa* (GCA_000182925), *Ophiostoma montium*, *O. piceae* (GCA_000410735), *O. novo-ulmi* (GCA_000317715) and *Sporothrix schenkii* (GCA_000474925) [33–35]. The search returned 158 candidate alignments from which primer pairs targeting *G. clavigera* were designed and tested on gDNA from cultures of *G. clavigera* and its the sister species *L. longiclavatum*, *L. terebrantis*, *L. wingfieldi* and *O. montium*.

For candidate alignments that successfully passed PCR specificity testing, two real-time TaqMan PCR probes were designed as follows: a probe used for the detection of gDNA was designed within the intron sequence located between the two primer pairs, whereas a probe targeting the cDNA was designed to span the exon-exon junction (Fig 1A and 1B). Primers were then optimized as recommended in Feau et al. [11]. To determine the real-time PCR efficiency for each primer pair and probe a five-point standard curve was developed over a range of 10-fold dilutions from 10ng/μl to 1pg/μl of gDNA.

## Wood inoculation

Eight isolates of *P. ramorum* (two from each phylogenetic lineage i.e. EU1, EU2, NA1 and NA2, [36]; Appendix 1) and one *G. clavigera* isolate were used for the wood inoculations. They were obtained from long term storage, plated on carrot agar [37] and 2% malt extract agar (MEA), and sub-cultured to fresh plates 10 days prior to inoculation. In order to simulate live infection on the host, living trees were freshly felled and prepared for the artificial inoculation. Three tree species were used for the *P. ramorum* inoculations: Douglas fir (*Pseudotsuga menziesii*), Japanese larch (*Larix kaempferi*) and Western hemlock (*Tsuga heterophylla*). Lodgepole pine (*Pinus contorta*) was used for *G. clavigera* inoculation. Logs (~12cm or greater in dbh) were brushed to get rid of excess debris, rinsed with water, cut into 0.5 m long bolts and inoculated with *P. ramorum*, *G. clavigera* or a blank agar plug used as a negative control as described in [36]. Inoculated bolts were misted with water and placed in plastic bags for 28 days.

## Heat treatment to determine mRNA stability

Pure cultures of *P. ramorum* and *G. clavigera* were subjected to two heat treatments: 1) a simulated spruce-pine-fir (SPF) kiln-drying schedule (from 15˚C to 70˚C for 7 hours) used to treat wood under the standards approved by the Canadian Food Inspection Agency (CFIA) [38]; and 2), exposure of the pathogens to 70˚C for 1 hour [38] (Fig 2). *Phytophthora ramorum*

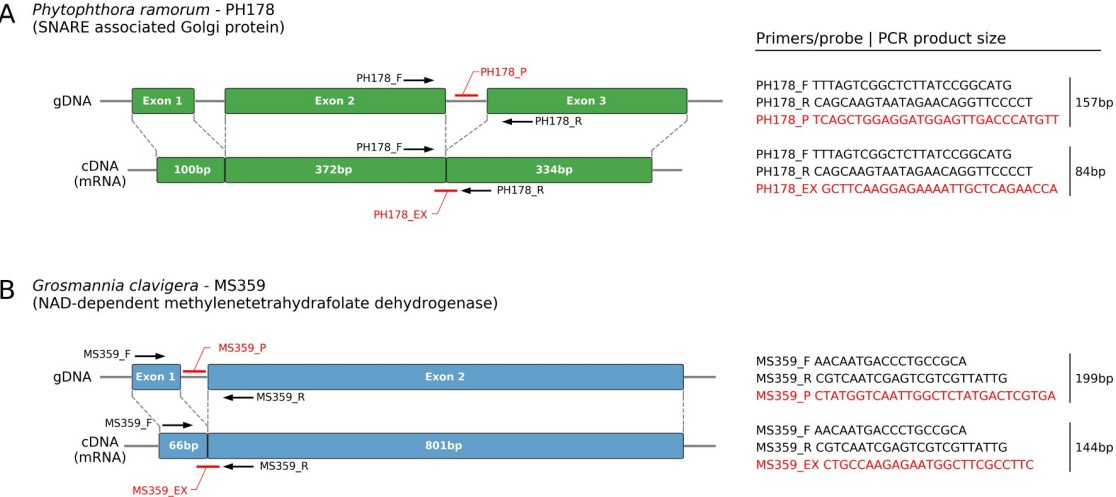

**Fig 1.** Real-time PCR detection assays targeting *Phytophthora ramorum* (A) and *Grosmannia clavigera* (B) gDNA and cDNA developed in this study.

isolate PFC-5073 [lineage NA2] and *G. clavigera* isolate KW1407 were grown on cellophane on V8 [39] and 2% MEA media (three replicates per time point). For the SPF kiln-drying schedule, mycelium was transferred from petri plates into 0.1mL PCR strip tubes and placed into a thermocycler. In parallel, one 1.5ml Eppendorf tube containing 30 mg of mycelia was immediately frozen to serve as a no-heat treatment control. The thermocycler was used to conduct a long heat treatment that simulated the kiln-drying schedule. Post-heat-treated samples were maintained at room temperature until its designated collection time point. The mycelium was sampled at 8 time points: 0, 6, 12, 24, 48, 96, 168 and 240 hours after the treatment. At

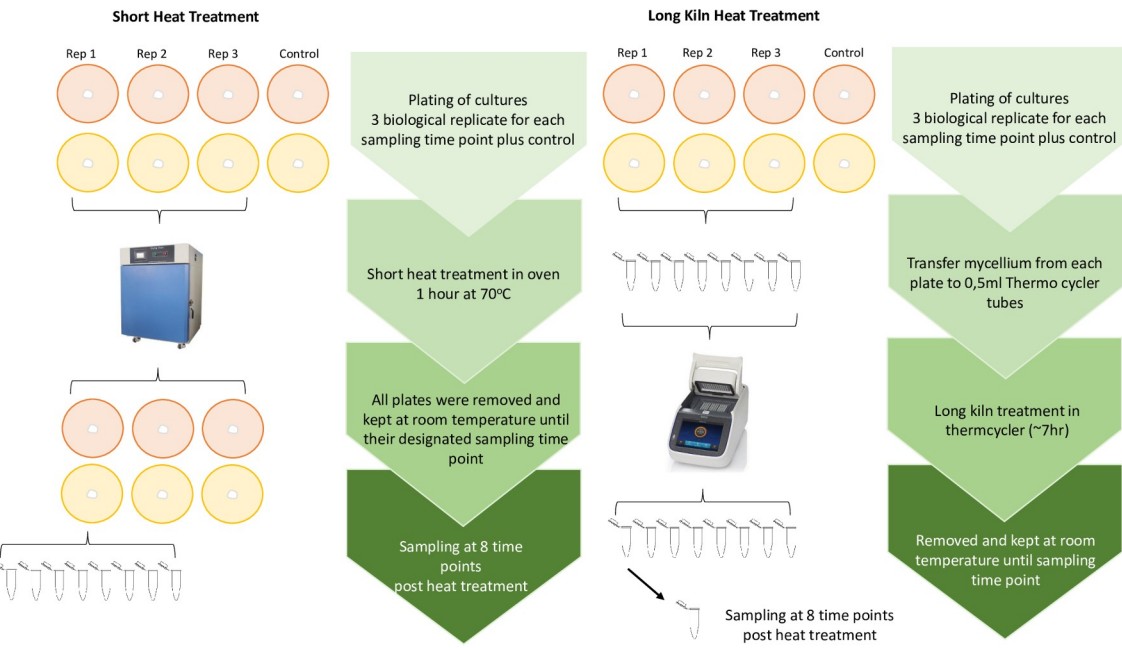

**Fig 2.** Work flow diagram of the short (left) and long kiln (right) heat treatments. Orange plates represent *Phytophthora ramorum* on V8 agar, while yellow plates are *Grosmannia clavigera* on malt extract agar (MEA).

each time point two mycelial samples were collected: one was transferred into a 1.5 ml micro-tube, submerged in liquid nitrogen and stored at -80˚C for subsequent DNA and RNA extractions; the other one was plated on three clarified V8 (*P. ramorum*) or MEA (*G. clavigera*) agar petri dishes and incubated in a dark growth chamber at room temperature for 28 days. For the short heat treatment, a laboratory oven was preheated to 70˚C to incubate 24 replicate plates of the *P. ramorum* and *G. clavigera* isolates for 1 hour. After this treatment, all plates were removed from the oven and placed at room temperature before collection at the time-points mentioned above. Mycelial samples were collected and stored as previously described for the SPF kiln-drying schedule.

## Chlamydospores collection

For the *P. ramorum* chlamydospore testing qPCR of gDNA (100 ng/µl) and RT-qPCR of cDNA (55ng/µl) were carried out for two different *P. ramorum* isolates (Pr-05-015 and CBS101329, both isolated from *Rhododendron* sp.) in biological duplicates and sample triplicates. Chlamydospores were harvested as described in Tsao [40] with the following modifications. Non-blended fungal mats were transferred onto a 75µm cell filter in a 50ml conical plastic centrifuge tube. Using sterile dH$_2$O, these mats were rinsed while being patted with a rubber policeman until the volume of water in the falcon tubes reached approximately 10–20 ml. This was repeated twice. The tubes were then centrifuged at 10,000 x g for 2 min, and the supernatant was removed. The chlamydospore suspensions were then aliquoted into several 1.7 ml micro-tubes and centrifuged at 8000 x g for 5 min removing the supernatant. The pellets were then resuspended, combined and centrifuged at 6,000 x g for 5 min and the supernatant discarded. Final concentration of the spores was adjusted to 1x10$^5$ chlamydospore/ml using a hemocytometer. The obtained spore suspension was centrifuged at 6000 x g for 5 min, discarding the supernatant. The pellets were then suspended to a minimum volume (approximately 50µl). The chlamydospore suspension was transferred to a Lysing Matrix C grinding tube (MP Biomedicals, Santa Ana, California, USA) followed by RNA/DNA extraction as it was done for mycelia from pure cultures.

## RNA and DNA extractions

Fifty to 80 mg of wood scrapings were collected at each inoculation point from eight logs of four conifer species and used for the simultaneous extractions of gDNA and mRNA. Wood samples inoculated with *G. clavigera* were placed in 15 ml vials with two 10 mm stainless steel balls and were submerged in liquid nitrogen to keep the samples frozen. Vials were then placed in the Geno/Grinder (SPEX SamplePrep 2010, Metuchen, New Jersey, USA) at 15,000 rpm for 30 seconds. Wood samples inoculated with *P. ramorum* were hand-ground using a mortar and pestle. Mycelial samples were placed in Lysing Matrix C. All samples were flash-frozen in liquid nitrogen, ground in a FastPrep-24 homogenizer (MPBiomedicals) at 5.5 rpm for 30 seconds and re-submerged in liquid nitrogen. For *G. clavigera*, samples were removed from the freezer and submerged in liquid nitrogen to inhibit RNA degradation. Samples were individually placed in a mortar, immersed in liquid nitrogen and ground up into fine powder.

Simultaneous extraction of gDNA and RNA was performed using the AllPrep DNA/RNA Micro kit (QIAGEN Inc., Valencia, CA) following manufacturer instructions. Three extractions were performed for each as replicates. Genomic DNA concentration was measured using the Qubit fluorometer and all culture samples were diluted down to 1ng/µl using nuclease-free water. The concentration of RNA samples was measured using the NanoDrop 1000 spectrophotometer before diluting down to 10ng/ul. RNA integrity was assessed by band fluorescence using an agarose gel stained with ethidium bromide (EtBr).

## cDNA synthesis, qPCR and RT-qPCR

Using the diluted RNA, cDNA synthesis was performed using the QuantiTect Reverse Transcription Kit (QIAGEN). Two μl of gDNA Wipeout buffer (7x), 10 ng of template RNA and a variable volume of RNase-free water for a total of 14 μl was used in the first step. Then 1 μl of Quantiscript Reverse Transcriptase, 4 μl of Quantiscript RT Buffer (5x) and 1 μl of RT primer mix was added to the gDNA eliminated solution.

Two TaqMan reactions per time point were performed for the measure of mRNA stability post heat treatment. This allowed differentiation of either gDNA or cDNA with their corresponding probes. Real-time PCR mix included 0.5X Quantifast Multiplex PCR MasterMix (QIAGEN), 400 mM of each forward and reverse primer, 20 mM of TaqMan probe and 2.2 ng of template (gDNA or cDNA) for a final volume of 10 μl. Thermal cycling parameters used were 5 minutes at 95˚C for enzyme activation, followed by 40 cycles of denaturation at 95˚C for 30 seconds and 60 seconds of annealing/extensions at 60˚C. The threshold was automatically set and generated with the Applied Biosystems StepOne™ software; qPCR efficiency was also calculated automatically through the standard curves tab.

## Statistical analysis

The proportion of viable pathogen was estimated by using the ratio of mRNA over gDNA quantity i.e. quantification cycle (Cq) ratio of the real-time PCR probe targeting cDNA over Cq value of the probe targeting gDNA. This ratio was the unit of measurement used to compare the efficacy of heat treatment. Using statistical analysis software (SAS 9.4), the significance of the treatment values were tested using a two-factor ANOVA split plot, where factor A is heat treatment and factor B is sampling time points after treatments.

# Results

## PCR assay design and performance

Five out of the six primer pairs targeting *P. ramorum* were eliminated because of the presence of detectable amplification products on the agarose gel when tested with *P. lateralis* DNA. The primer pair that was selected for the *P. ramorum* assay (PH178) targets portions of a gene (Protein ID 74159; [23]) encoding for a predicted SNARE associated Golgi protein (Fig 1A) and yielded amplification products only with gDNA of *P. ramorum*. This gene was highly expressed in two mRNA profiling experiments with mycelial colonies of *P. ramorum* growing on agar media [28,29].

All 11 primer pairs designed to target *G. clavigera* amplified DNA from at least one non-target species. The primer pair targeting gene MS359 yielded a PCR product with all *G. clavigera* isolates tested and the closely related species *L. longiclavatum* (99.7% similarity with *G. clavigera* in the ribosomal internal transcribed spacer and 97.5% similarity at the genome level). Since these sister species occupy a similar niche (mountain pine beetle galleries) and have similar biology [30], we selected the assay targeting this gene to develop the mRNA assay (Fig 1B). MS359 (= *G. clavigera* GLEAN_5973; [27]) encodes for a putative NAD-dependent methylene-tetrahydrafolate dehydrogenase that is involved in the glyoxylate and dicarboxylate metabolism pathway. This gene is known to be expressed in growing mycelium and non-germinated spores and showed stable expression levels in RNAseq experiments at 12 and 36 h post-inoculation on media supplemented with host-defense metabolites and untreated control media [27,41]. Both testing genes were chosen based on their assumed functions related to basic metabolic processes and their high expression levels in transcriptome analyses [27–30].

We verified the presence of an intron in the gDNA samples by comparing the size of the PCR products obtained by amplification of the gDNA and cDNA reverse-transcribed from mRNA of the same samples. A smaller amplification product was obtained in the PCR of the cDNA (84 bp) than the gDNA (157 bp) of gene PH178, confirming the presence of the intron in the gDNA of *P. ramorum* (Fig 1A). Similarly, we observed a difference in amplicon product sizes between cDNA (149 bp) and gDNA (199 bp) in MS359 of *G. clavigera* (Fig 1B).

TaqMan probes were added to the two selected primer pairs to design real-time PCR assays that were tested for amplification efficiency on serial dilutions of gDNA. For the assay PH178 targeting *P. ramorum*, the standard curve yielded a regression coefficient of 0.998 indicating low variability between independent DNA isolations and an amplification efficiency of 92.3% (Fig 3A). Efficiency of the MS359 assay targeting *G. clavigera* was higher (100.3%), and variability slightly lower with a regression coefficient of 0.981 (Fig 3B).

## Assessing the performance of the detection assays in infected wood and chlamydospores

For each combination of isolate x tree species tested a necrotic lesion was observed around the point where the microorganism was inoculated, confirming the growth of these pathogens in wood. No similar necrotic lesion was observed in the controls (Fig 4). Genomic-DNA and cDNA obtained from these inoculated wood samples was successfully detected by their respective real-time PCR assay with Cq values ranging from 24.13 (*G. clavigera* inoculated on *P. contorta*) to 29.4 (*P. ramorum* inoculated on *L. kampferi*) for gDNAs (mean = 26.6 ±2.21). The Cq values obtained for the cDNAs were slightly higher, ranging from 27.4 (*G. clavigera* x *P. contorta*) to 33.4 (*P. ramorum* x *L. kampferi*) with an average of 31.2 (±2.77), indicating later detections than for gDNA (Fig 4).

We tested the efficiency of the PH178 assay on nucleic acids extracted from *P. ramorum* chlamydospores. The mean Cq was 25.45 for gDNA (± 0.055) and 29.05 for cDNA (±0.038) (data not shown). No amplification was observed with gDNA and cDNA from non-inoculated wood samples as-well-as the no-template controls.

## Use of molecular assays to assess viability of pathogens following heat treatment

Ratio of mRNA over gDNA quantity as measured by $C_q$ values for cDNA and gDNA were compared following two heat treatments (SPF Kiln-drying and short heat treatment) applied to mycelium of *P. ramorum* and *G. clavigera*. For both pathogens, we found a highly significant effect of the heat treatment (F = 72.4, $P < 0.0001$ and F = 25.2, $P < 0.0001$ for *P. ramorum* and *G. clavigera*, respectively) and an interaction between treatment and time point at which the gDNA and mRNA samples were collected (F = 4.2, $P < 0.001$ for *P. ramorum* and F = 2.7, $P < 0.01$ for *G. clavigera*). This likely resulted from the difference in cDNA detection as for both species gDNA was amplified after each treatment with $C_q$ values similar or higher than those obtained with the no-treatment control (e.g. $C_q$ distribution ranging from 24.0 to 34.0 for the two heat treatments versus 24.0 for the controls; Fig 5A, 5B, 5E and 5F). cDNA was detected for either pathogen at the end of SPF kiln-drying schedule treatment ($C_q$ value equal or above 40.0; Fig 5D and 5H), suggesting that this treatment efficiently killed the two pathogens. In contrast, the short heat treatment at 70˚C for 1 hour seemed to be less efficient than the longer kiln treatment, with clear evidence that the mRNA degraded at different rates for the two organisms following treatment. For *P. ramorum*, cDNA of the targeted gene (PH178) was detected after up to 24 hours post-treatment (mean $C_q$ = 35.8 ±4.82 for 0 to 24 hours post-treatment; Fig 5C). Similarly, *G. clavigera* cDNA of gene MS359 was detected until 96 hours

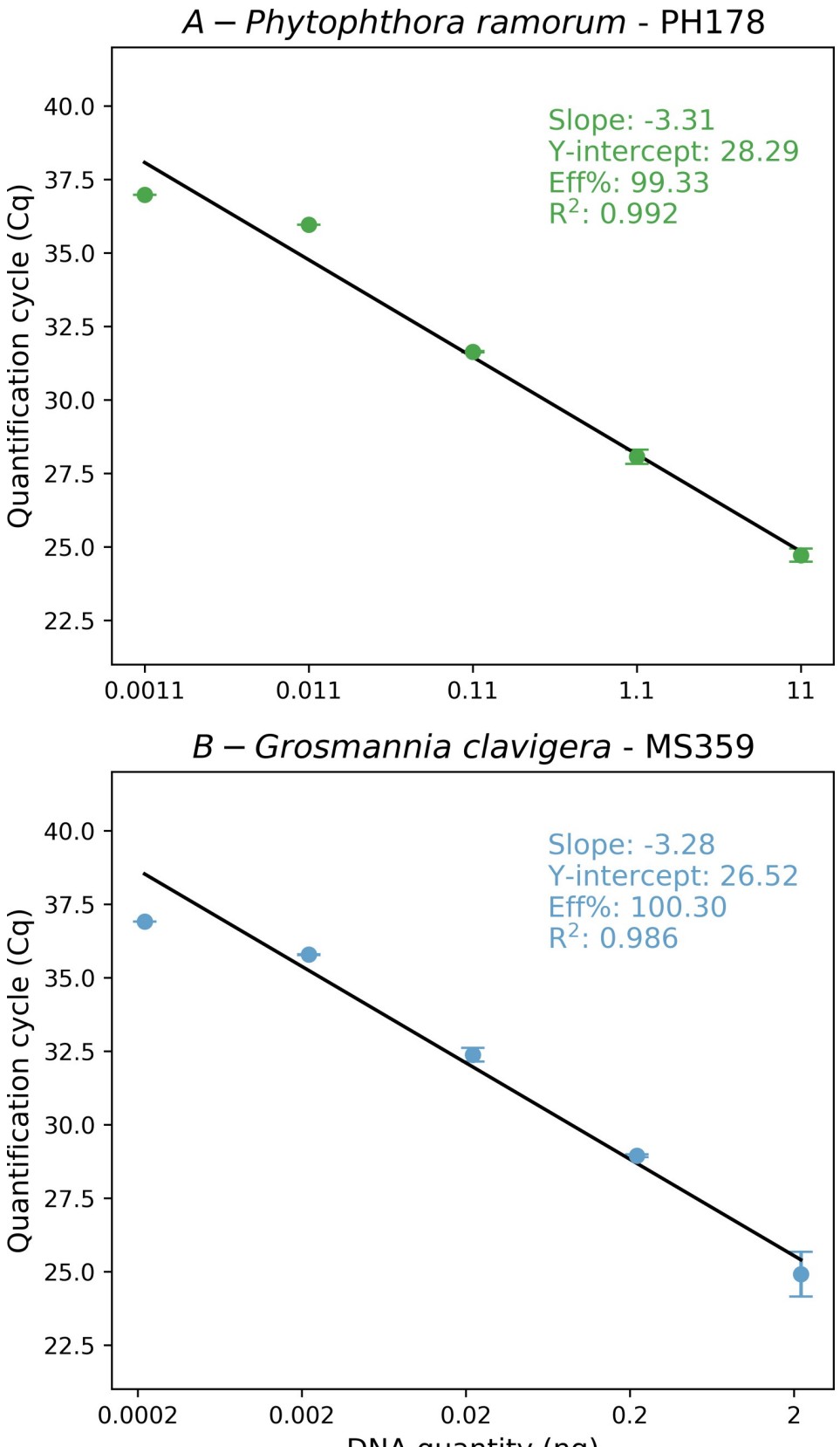

**Fig 3.** Log-transformed standard curve assessed with gDNA serial dilution (1:10) of (A) *Phytophthora ramorum* for the TaqMan probe PH178_EX ($R^2$ = 0.992 Eff% = 99.330) and (B) *Grosmannia clavigera* for the TaqMan probe MS359_EX. ($R^2$ = 0.981 Eff% = 100.303).

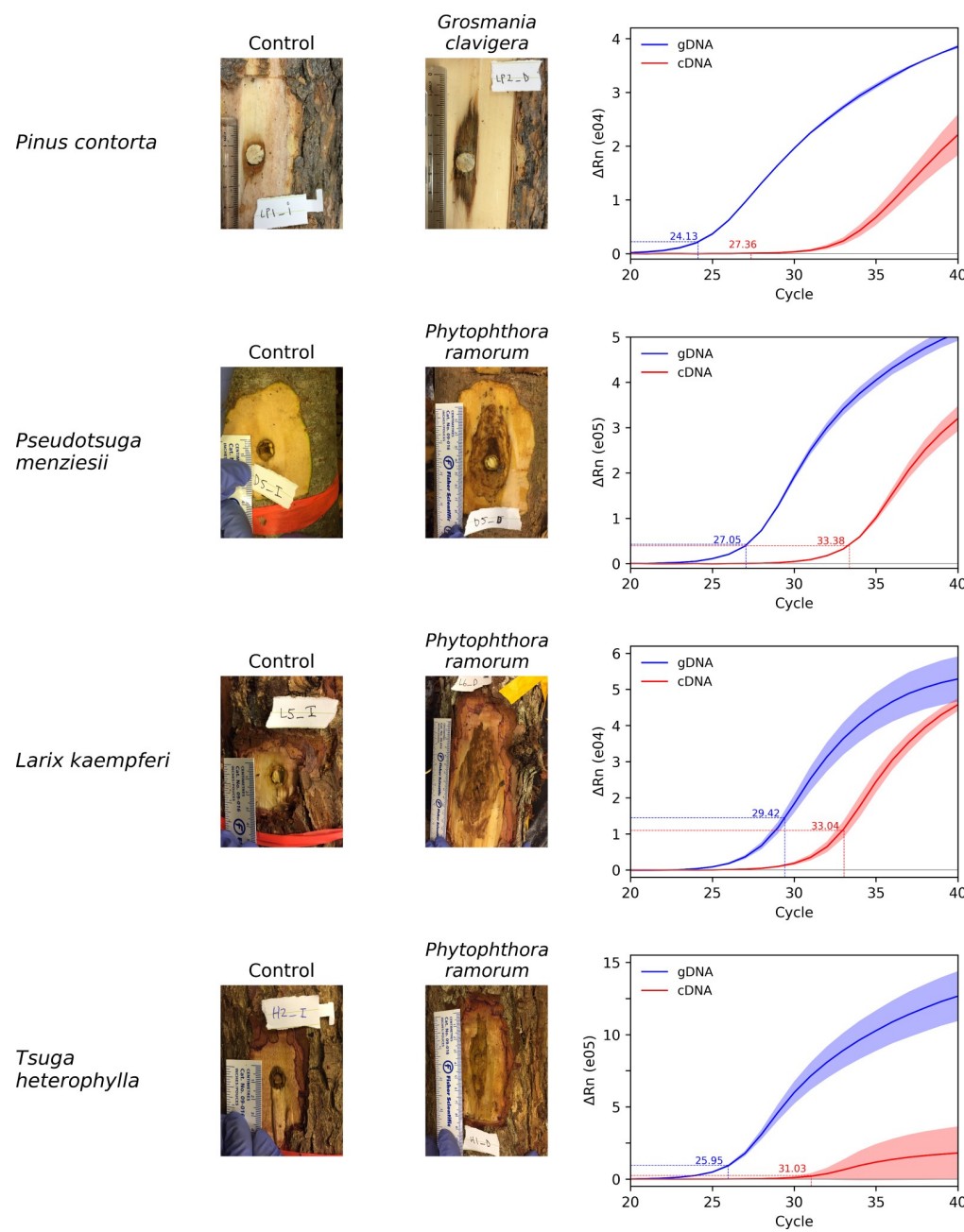

**Fig 4. Wood inoculation with *Phytophthora ramorum* (EU2 isolate P2111) and *Grosmannia clavigera* (isolate KW140) and corresponding real-time PCR amplification plots.** For each wood inoculation sample, gDNA and cDNA synthetized from mRNA extracted from a lesion 28 days post inoculation was tested in real-time PCR with either the PH178 assay (targeting *P. ramorum*) or MS356 (*G. clavigera*). Quantification cycle (Cq) values for gDNA (blue) and cDNA (red) are reported on each graph. Shaded area around the average line represents ±SD.

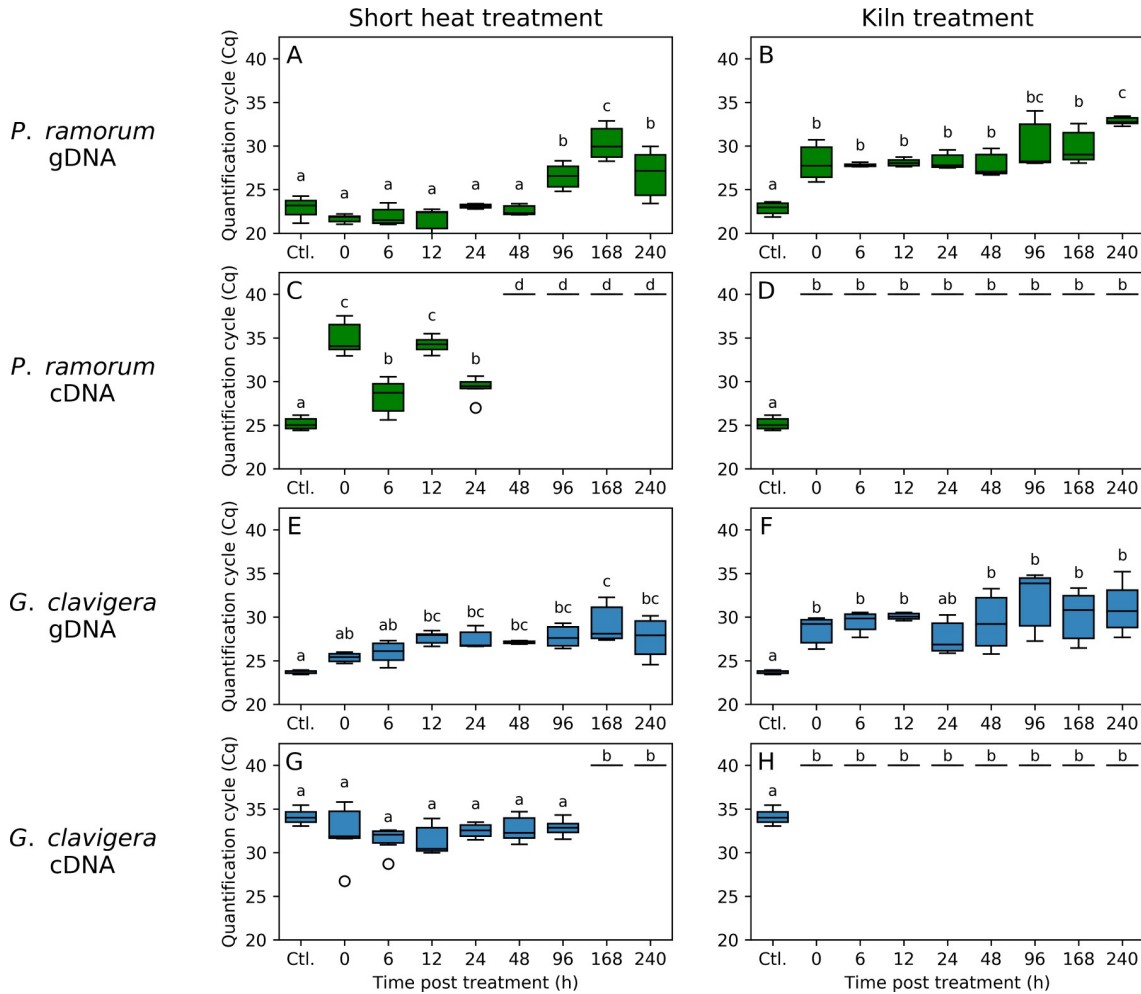

**Fig 5.** Efficacy of the short heat and SPF kiln-drying treatment of *Phytophthora ramorum* (green) and *Grosmania clavigera* (blue). Each graph represents distributions of cycle-threshold ($C_t$) values obtained by real-time PCR with the PH178 assays (targeting *P. ramorum*) or the MS356 assays (*G. clavigera*) for gDNA or cDNA extracted from cultures sampled from 0 to 240 hours after treatment. Boxplot means sharing a letter are not significantly different ($p > 0.05$) according to a Tukey's HSD test.

after the short heat treatment (mean $C_q$ = 34.2 ±3.75), suggesting incomplete degradation of mRNA and/or non-lethality of this treatment during this time (Fig 5G).

## Assessment of pathogen viability after heat treatments

Small samples of mycelia were collected at time point zero just after each heat treatment and plated onto a sterile clarified-V8/MEA agar Petri dish and grown for 28 days. No growth was observed from any of the cultures, suggesting that the two heat treatments had been lethal.

## Discussion

Quantitative real-time PCR is increasingly becoming the method of choice for molecular detection of microorganisms. Conventional methods used for assessing the efficacy of treatment of wood products typically involve culturing the microorganisms. These methods have the advantage of simultaneously assessing the presence of microorganisms and their viability, but require days from initiation to result and can produce a high rate of false negatives. The

present study was undertaken to evaluate the feasibility of developing a molecular method for assessing the presence and viability of wood-colonizing pathogens. Using qPCR and RTqPCR with TaqMan probes that overlap the exon-intron and exon-exon junctions, it was possible to distinguish the mRNA and the corresponding genomic gene copy (gDNA). This allowed comparison of the kinetics of degradation of both molecules following heat treatments and provided a measure of the presence and absence of microorganisms as well as an assessment of their viability.

Accurate detection of invasive pathogens is a hallmark of efficient prevention and integrated pest management programs [42,43]. In some instance assessing pathogen viability may even be more crucial. This is particularly relevant in the context of plants, plant parts and wood-material trade between countries [1–3]. RNA-based molecular assays have proven to be successful in detecting a number of different pathogens and assess their viability. For example, early studies have used levels of ribosomal RNA (rRNA) as a proxy of bacteria viability involved in periprosthetic joint infections [44], human endophthalmitis [45] and clinical infectious diseases [46]. Although all these studies reported correlation between rRNA detection signal and cell death, rRNA half-life and inconsistent retention after cell death makes it somewhat less accurate, particularly for short term experimentation [46]. Because of its relatively short half-life, mRNA has been used successfully as a viability indicator for a number of prokaryote [46–48] and eukaryote pathogens [49,50]. Another drawback of rRNA for assessing cell viability is that it does not have spliced introns and therefore it cannot be used to differentiate between RNA and gDNA templates, raising the possibility of false positive amplification from accidental gDNA contamination. In this case, a complete elimination of DNA is required prior performing RT-qPCR to ensure reliable use of rRNA as proxy of cell death. As an alternative method to differentiate between RNA and gDNA templates, Menzel et al. designed an end-primer which spans on the exon-exon junction in mRNA, so that efficient primer-annealing and PCR-amplification is only possible after splicing the intron of the targeted gene during the DNA-translation process [51]. Similarly, Leal et al. developed several reverse transcription real time PCR assay to detect living the pinewood nematode *Bursaphelenchus xylophilus* in wood by targeting the presence of mRNA [17,52]. These studies showed the potential of mRNA for routine detection of living pinewood nematode in commercially manufactured wood-pellets for living pinewood nematode [53].

*Phytophthora ramorum* causes sudden oak death in North-America and sudden larch death in UK. Despite the regulations that are in place to prevent movement of this pathogen in North America, Europe and Asia *P. ramorum* has spread into new areas through nursery stock exchange [54]. Accurate and rapid detection methods are crucial for preventing new introductions of this pathogen and several molecular assays have been developed to improve its detection and monitoring [31,55–57]. *Phytophthora ramorum* is also considered as a wood-colonizing pathogen as it can grow in xylem vessels underlying phloem lesions on several woody-perennial species [58,59]. In this context, the ability to assess its viability in plants, plant-parts and in wood-derived products following treatment may be necessary. Chimento et al. [60] developed real-time PCR primers targeting the cytochrome oxidase subunit I (COX 1) gene to investigate the viability of *P. ramorum* mycelial cultures after different treatments [60]. As COX1 is a mitochondrial gene that does not contain introns in *Phytophthora* spp. [61], their real-time PCR assay does not discriminate the genomic of the transcribed gene copies. In this study the specificity of the assay to detect living *P. ramorum* was increased by amplifying exclusively cDNA, ruling out potential false positives generated by gDNA contamination of the RNA sample.

The second assay we developed targeted *G. clavigera*, one of the mountain pine beetle fungal associate. This wood-colonizing organism was selected as a second case-study to

demonstrate the efficiency of our approach in assessing the viability of an infectious fungus. Due to the high sequence similarity between *G. clavigera* and the closely related species *L. longiclavatum* [62,63], the assay developed amplified mRNA and gDNA of both species. This could be a positive attribute of this assay since both species are symbionts of the mountain pine beetle and share a very similar ecological niche in pine trees [22]; in fact it is possible that these fungi hybridize and the species limits are not clear (R. C. Hamelin and A. Capron, unpublished). These fungi cause only a wood discoloration symptom ("blue stain") and no structural damage to the infected wood. The only issue with regard to the trade of wood products is related to the risk of long distance spread and introduction of these fungi in areas where they do not occur [3]. Since these fungi require bark beetles for dissemination, they pose a limited threat.

Our aim was to use our gDNA and mRNA-targeted real-time PCR detection assays to assess the viability of two wood-infecting microorganisms following lethal treatments to test their efficacy. The underlying hypothesis tested was that viability is related to the expression of specific genes. Therefore, monitoring specific *P. ramorum* and *G. clavigera* gene transcripts by real-time PCR was expected to provide a simple proxy for viability of these two organisms. The choice of transcript was an important consideration, not only for sensitivity but also for its expression under a variety of conditions. Genes selected to assess viability should be constitutively expressed regardless of the environmental conditions and the micro-organism lifestage as opposed to those induced following a specific environment signal [64]. This ensures that the lack of expression of a constitutive gene is due to the death of the organism instead of the gene being turned "off" by a specific environmental factor. Therefore, in order to develop markers as indicators of cell viability, it was important to identify transcripts that are present and expressed at different stage of the organism's life cycle and under different environmental conditions. The transcripts retained for this study were chosen based on their putative function related to basic metabolic processes and their high expression levels in transcriptome analyses [27–29,41]. In addition, we validated their expression in pure cultures and for mycelium growing in wood-logs. For *P. ramorum* we successfully conducted the viability assay on chlamydospores, the asexual reproductive structures that are generally involved in long-term survival under adverse conditions. Although there are no reports of chlamydospores surviving in wood, they could be present in other tissues such as the bark, roots or soil associated with roots or leaves [65].

Several studies showed that the ISPM No. 15 guidelines for treatment of wood packaging material [4] were not always lethal for wood-colonizing pathogens. For example, the protocol requiring exposure for 30 minutes at a temperature of 56˚C was lethal for *P. cinnamomi* and some other species, but not all the wood colonizing species tested [66]. No mycelial growth was observed for *P. ramorum* in tanoak disks and boards after a heat treatment of 60˚C for at least one hour [67]. However, Chimento et al. demonstrated that a 60˚C treatment for one hour did not kill the pathogen but instead delayed its growth [60]. Based on these discrepancies and since most kiln and heat treatments generate core temperatures in wood products exceeding ISPM15 requirements (e.g. for SPF products, drying temperatures are usually in the range of 70˚C to 80˚C; [68]), we applied on mycelial cultures a treatment of 70˚C for a minimum of one hour and the kiln-drying treatment that requires temperatures from 15 to 70˚C for 7 hours approved as a standard to treat SPF approved by the CFIA [38]. Accurate detection of the cell viability is therefore crucial in investigating the efficacy of these heat treatments. We used two different methods to assess cell viability. As each of these methods is based on criteria that reflect different levels of cellular integrity or functionality their outcome in declaring cells alive or dead can be different [48,69]. A classical view is to consider that viable cells still have the potential to multiply under suitable condition [48]. This was assessed in our study through

a culture-based method. Though this method has a high rate of false negatives [70,71] and several studies have demonstrated that cells that have lost their "culturability" following a lethal-treatment may still retain some of their physical integrity and functionality [69,72,73]. As a second method, we assumed that the rapid degradation and synthesis inhibition of mRNA might be a useful indicator of cell mortality [48]. However, because mRNA was still detected by reverse transcriptase real-time PCR in *P. ramorum* and *G. clavigera* dead cells (as indicated by the non-recovery of living cultures) several hours following a short heat treatment (70˚C for a minimum of one hour), we couldn't conclude that mRNA provided an absolute indicator of viability according to this assumption. Instead, we can consider that the kinetics of mRNA disappearance is related to loss of cell viability: we observed that passed a certain time-point after the heat treatment (i.e. 48 hours for *P. ramorum* and 168 hours for *G. clavigera*), mRNA was no longer detected. Non-detection of mRNA just after the SPF kiln-drying schedule treatment supports this hypothesis. Possibly, the longer time of exposure to lethal temperatures in this schedule (7 hours) contributed to achieve full mRNA degradation for that treatment.

The approach presented in this study could be improved by technical improvements before envisaging operational use. For each combination of pathogen x tree tested, detection of the cDNA was always delayed comparatively to gDNA; in few cases we observed a substantial Cq value difference between the gDNA and cDNA (e.g. *P. menziesii* and *T. heterophylla*; Fig 4). Extraction of RNA from plant tissues is notoriously challenging. Particularly, yield and quality of the RNA purified from wood, in particular from conifers, is limited by the high content in polyphenols, polysaccharides and other secondary metabolites [74,75]. These compounds tend to co-precipitate with the RNA in the presence of alcohols, thereby remaining in the final extract, resulting in RNA instability and interfering with downstream enzymatic reactions such as reverse transcription and cDNA synthesis [76,77]. Optimization of RNA-extraction protocol should help improve cDNA synthesis and therefore sensitivity of the cDNA-assays.

In addition, a Two-step RT-qPCR was used for these experiments, as we thought this approach would be more convenient for a proof-of-concept study with cDNA detection. Although this approach provides more flexibility and control than a one-step RT-qPCR, it takes more time to implement and can introduce more errors. A one-step RT-qPCR involves the reverse transcription and the PCR reaction within one single tube and would be more ideal for high throughput screening. However, this method also requires careful evaluation of the conditions for both the cDNA and PCR steps that may not be optimal for either reaction. Downstream optimization of the current assays using a one-step approach should be considered but was beyond the focus of the current experiment. Another consideration would be the use of the DNA-binding photoreactive dye Propodium monoazide (PMA) that enables to distinguish between dead and living cells [78]. However, despite the ease and success in use with some organisms, the greatest concern is its limitation in technical use on environmental samples [79].

The Food and Agriculture Organization (FAO) often revises the list of suggested treatments for wood packaging material, emphasizing the importance of reducing the risk of quarantine pests associated with wood exports [4]. The present research acts as a proof-of-concept to enhance future molecular detection methods of invasive and native pathogens of phytosanitary concern. Our mRNA/gDNA detection method differentiated between dead and alive pathogens using reverse transcriptase and real-time PCR assays and was validated in pathogen-infected wood samples. These assays should provide a novel way to evaluate and compare the efficacy of wood treatments to eliminate microorganisms that are difficult to detect. Further investigation of the timing and pattern of mRNA degradation in wood and other plant tissues such as leaves and bark should be conducted using this method.

## Acknowledgments

The authors would like to acknowledge the technical help of Brett Ford and Julie Sheppard. Logs for the inoculations were provided by The University of British Columbia Malcolm Knapp Research Forest.

## Author Contributions

**Conceptualization:** Barbara Wong, Isabel Leal, Nicolas Feau, Angela Dale, Adnan Uzunovic, Richard C. Hamelin.

**Formal analysis:** Barbara Wong, Isabel Leal, Nicolas Feau, Angela Dale.

**Funding acquisition:** Isabel Leal, Adnan Uzunovic, Richard C. Hamelin.

**Methodology:** Barbara Wong, Isabel Leal, Richard C. Hamelin.

**Supervision:** Isabel Leal, Nicolas Feau, Angela Dale, Adnan Uzunovic, Richard C. Hamelin.

**Visualization:** Nicolas Feau.

**Writing – original draft:** Barbara Wong, Isabel Leal, Nicolas Feau, Richard C. Hamelin.

**Writing – review & editing:** Barbara Wong, Isabel Leal, Nicolas Feau, Angela Dale, Adnan Uzunovic, Richard C. Hamelin.

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
