## [Decision Letter · Decision Letter 0]

30 Aug 2019

PONE-D-19-22766

Molecular assays to detect the presence and viability of Phytophthora ramorum and Grosmannia clavigera

PLOS ONE

Dear Dr Hamelin,

Thank you for submitting your manuscript to PLOS ONE. After careful consideration, we feel that it has merit but does not fully meet PLOS ONE’s publication criteria as it currently stands. Therefore, we invite you to submit a revised version of the manuscript that addresses the points raised during the review process.

We would appreciate receiving your revised manuscript by Oct 14 2019 11:59PM. To enhance the reproducibility of your results, we recommend that if applicable you deposit your laboratory protocols in protocols.io, where a protocol can be assigned its own identifier (DOI) such that it can be cited independently in the future. For instructions see: http://journals.plos.org/plosone/s/submission-guidelines#loc-laboratory-protocols

We look forward to receiving your revised manuscript.

Kind regards,

Simon Francis Shamoun, Ph.D.

Academic Editor

PLOS ONE

Journal Requirements:

Grant from Genome Canada (10106) to RCH

Grant from the Genomic Research and Development (GRDI) initiative to IL

We note that one or more of the authors are employed by a commercial company: FPInnovations

Additional Editor Comments:

Dr. Richard Hamelin

Professor, UBC- Vancouver, BC, Canada

Bonjour Richard,

Your manuscript Number PONE-D-19-22766:

Molecular assays to detect the presence and viability of Phytophthora ramorum and Grosmannia clavigera, was critically reviewed by 5 external reviewers. Based on these reviews and my own assessment, I recommend publishing your manuscript after "Major Revision".

I would like to draw your attention to address the comments/suggestions made by the reviewers. In particular, please, pay a close attention to reviews of the reviewers #1 and #5.

I would like to review your revised version of your manuscript before considering it for publication. You have 45 days from August 30, 2019 to submit your revised version of the manuscript to PLOS ONE.

Thank you for submitting your research results to PLOS ONE. We look forward to receiving your future research manuscripts.

Best regards,

Simon

Simon Francis Shamoun, Ph.D

Academic Editor

PLOS ONE

Reviewers' comments:

Reviewer's Responses to Questions

**Comments to the Author**

1. Is the manuscript technically sound, and do the data support the conclusions?

Reviewer #1: Partly

Reviewer #2: Yes

Reviewer #3: Yes

Reviewer #4: Yes

Reviewer #5: No

2. Has the statistical analysis been performed appropriately and rigorously? 

Reviewer #1: Yes

Reviewer #2: Yes

Reviewer #3: Yes

Reviewer #4: Yes

Reviewer #5: N/A

3. Have the authors made all data underlying the findings in their manuscript fully available?

Reviewer #1: Yes

Reviewer #2: Yes

Reviewer #3: Yes

Reviewer #4: Yes

Reviewer #5: Yes

4. Is the manuscript presented in an intelligible fashion and written in standard English?

Reviewer #1: Yes

Reviewer #2: Yes

Reviewer #3: Yes

Reviewer #4: Yes

Reviewer #5: Yes

5. Review Comments to the Author

Reviewer #1: This manuscript describes the design of molecular tools for the specific detection of genomic DNA and messenger RNA from two plant pathogens, i.e. the oomycete Phytophthora ramorum and the fungus Grosmannia clavigera. The novelty brought by this work is the design of species-specific tools on regions covering exon-intro junctions; in such a way that only cDNA retrotranscripted from mRNA is made amplifiable by PCR. The authors compared the amplification of coding regions of gDNA and corresponding mRNA in real-time PCR in several experimental trials, and showed that the level of mRNA decreased after mycelium treatment (e.g. heating) and could be used as a proxy for assessment of fungal or oomycete viability.

The manuscript is well written, very easy to follow and overall well organized (although I suggest minor changes, see below). The discussion section is exhaustive and interesting, although more light is shed on P. ramorum.

The experiments are adequate and compelling for most of the work, but I noticed a major discrepancy between what is claimed to be done and what is reported in the mat&met and results sections. In this respect, I recommend a major revision.

Indeed, the abstract states that “a stability analysis was conducted by comparing the ratio of mRNA to gDNA overtime following heat treatment of wood infected by the oomycete..”. Also P12L244, it is written: “Ratio of ….applied to WOOD-LOGS infected with P. ramorum and G. clavigera”. However, in the experiments reported in the manuscript, only mRNA and gDNA from MYCELIAL cultures of the pathogen were heat-treated, not from WOOD infected by the latter (see §Heat treatment to determine mRNA stability”, P6-7; in Figure 4 caption “… for gDNA or cDNA extracted from CULTURES sampled from 0 to 240 hours after treatment.”), and in discussion section L285-287, L382.

Likewise, all the discussion parts related to the appropriateness of wood treatments deserve mitigation since the experiments carried out in this work only dealt with in vitro-cultivated mycelium and not in vivo growing mycelium. The efficiency of heat treatment procedures are dependent of wood thickness, relative humidity, heat flow, etc.

I therefore suggest clarifying what was really assessed in this work. It seems that artificially infected wood were indeed tested after 28 days of incubation, but not after the two different heat treatments.

Specific comments:

L46: Ref 8 and 9 do not seem to be the most relevant ones to support the statement of the sentence.

L50: again here, ref 11 does not seem to be appropriate to the support the statement.

L69: ref 21 and 22 are more appropriate here, ref 20 a bit less supporting. Consider removing it.

L79 and 90: please could you explain why distinct DB were used for the selection of genes?

L82: on which basis the genes were selected? It is written “genes are expected to be expressed”, but how was that anticipated? The discussion section contains important information pertaining to the rationale followed for the selection, and this information should be moved to the mat&met section.

L133: number, reference of isolates and number of replicates should better be indicated here, rather than in a later paragraph (L177-179).

L140-141: the sentence sounds a bit odd.

L147: please specify what the wood samples were collected from.

L177: how was the qPCR done with these solutions as templates? 2.2 ng of template DNA are indicated a few lines above. Is it still the case?

Figure 1 caption indicates that gDNA and cDNA were used, but it is in conflict with what is written L215-216.

L201: is there one single sister species (L. longiclavatum) or several involved here? Not clear what “these sister species” refer to.

Figure 3 should remind how many replicates were included, and also explain how the curves should be interpreted (shaded areas, min and max?). Also, it should be interesting to discuss the striking lower magnitude of fluorescence yielded with cDNA extracted from P. ramorum-infected Tsuga, whereas the gDNA curves look “normal”. I noticed that there was a change of y-axis scale compared to the other plots, probably in order to better fit the gDNA fluorescence magnitude. But still, the gDNA/cDNA difference with the other plots seems awkward.

L239: I do not think that it is relevant to compare the mean Ct values obtained with chlamydospores to those obtained with infected wood samples. Chlamydospores quantity was arbitrarily set, we do not know if is close to reality in infected tissue or soil for instance.

Table 1: consider removing it, as it is a piece of arid stats. Or maybe replace “treatment” with what it stands for.

L285: heat treatment may just not degrade mRNA, it may also inhibit its synthesis.

L367-L372: I fully agree with this caveat. The authors should therefore recommend a practical procedure to assess the kinetics of mRNA.

Figure 4: The authors should discuss some questioning variations observed in cDNA quantity, e.g. the high variation in cDNA amplification between Ctrl/ 0 and 24 hrs post treatment for P. ramorum/short heat treatment. Although it seems obvious that after a certain treatment duration, no mRNA is produced, what would explain this high variation before that time? One may also wonder why gDNA and cDNA quantities (inferred from Ct values) in the control are sometimes significantly superior to the 0 tpt values? In two occasions, no mRNA is detected at all for 0 tpt, whereas the ctrl value is “normal”. Is it because 0 means that the treatment already started? If yes, 0 is propably not appropriate term.

More general comments:

The discussion section is a lot about P. ramorum, and unfortunately G. clavigera is a bit set aside. Although G. clavigera is less “famous” than P. ramorum, a more balanced discussion would be nice.

Reviewer #2: This manurscript is very appropriate for PLOS ONE. The manuscript is well written and present a new molecular assay using mRNA and gDNA to detect and show viability of Phytophthora ramorum and Grosmania clavigera. The method is well described and present new material. The 2 assays seem to work very well and could serve as base for ID and evaluation of viability of forest pathogens. The genomic was used to ID markers and selection of junction of intron/exon was good choice and well presented. The discussion could have more information comparing other methods and talk about the limitation of the assays, by example the detection of mRNA is very at the Ct value of limit of detection. What about other assays published. Other missing or clarifying information need to be add.

Specific comments:

P4L58 did you consider to add other method information to look at viability, I believe the PMA Propodium monoazide is a good method and do not need to play with RNA. Should add info and pro and cons in discussion?

P4L69 I was wondering about if the P. ramorum was infecting in wood or surface but was more clear later in manuscript. Could be presented more here. P5L82, it is really a communication or more a resource? (Bret Tyler sequences)

P6L104 for G. clavigera, only one isolate was use, how do we know it would work with multiple genotypes?

P6L109 any reason on use of only conifers, what about deciduous tree for P. ramorum?

P6L115 any specific reference for the SPF kiln drying schedule?

P8L144 "Matric" should be "Matrix" Correct in line 154.

P9L178 "586 and 101329) info on specimens, it is voucher isolate, CBS culture??? need more info on you isolates used in the manuscript?

P9 Just wondering it is better ration Ct or Ration concentration calculated, Ct,Log10?

P12L232, 33.4 seem to be very low limit? not so high you may miss lower concentration of pathogens. also should explain the range in figure 3, blurry red?? Blurry blue, same Ct but range blurry?? not clear what it mean?

P12L247 and Table 1, not so clear what the F, Fvalue is the stat need to be better explained and significance in the manuscript?

Table 1 no footnotes, what the "*" means? By? Tyope III SS?

Discussion, RNA extraction and cDNA is additional step, add info how the RNA extraction is easy or difficult and cDNA prep, risk of contamination, compare to PMA should be discuss.

What is the Ct level limit for cDNA-mRNA in other publication using similar approach?

Reviewer #3: In this MS, development of a PCR-based viability assay of tree pathogens is described. The authors showed that both genomic DNA and mRNA extracted from tree tissues under the bark were amenable for qPCR. They also showed that heat treatments could effectively kill the pathogens. Effectiveness of the heat treatments on the pathogens growing in the wood trunk was, however, not evaluated. The author needs to address this point. Below are my specific comments.

Line 18: RNA represents ... therefore only be produced by living organisms.

<comment> RNA, as well as DNA, are only produced by living organisms.

Line 164: cDNA synthesis

<comment> The authors used two-step RT-qPCR for all samples. The two-step method, which uses a stock cDNA, may be advantageous for the screening of primer pairs. However, for high-throughput assay, one-step RT-qPCR is more advantageous and give less experimental variation.

Line 231: Cycle-threshold values obtained for the cDNA were slightly higher.

<comment> For both P. ramorum and G. clavigera, the differences of Ct values between gDNA and cDNA were over 3, which were equivalent to an 8-fold difference in copy number of templates. This is a huge difference and needs to be discussed.

Line 242: No amplification was observed in the no-template control.

<comment> The vast quantity of genomic DNA and cDNA from inoculated wood samples were of the host plant origin. What was the level of background amplification when a DNA template from a non-inoculated plant tissue was used?

Line 342: indicating that the target gene is still expressed during this life-stage.

<comment> Most likely, the target gene is not expressed in the resting spores, but its transcript is stored in them.

Reviewer #4: This is an interesting manuscript describing a novel approach to the development of real-time PCR assays for viable pathogens in wood, providing significant improvements to current methods used in screening wood products. In particular the focus on only detecting viable pathogens (linked to the decay of mRNA) is valuable and will likely be applied in plant diagnostics. The manuscript is very well written, relatively free of grammatical and spelling errors (see short list below), and the statistical analyses appear sound.

The real time PCR data depicted in Fig 3 is however a bit troubling, with regard to the Ct values. The authors should mention the instrument used in the assays in the Methods section, L 170-176, and how the thresholds were established for Ct calculations. A more precise estimation of the Ct could improve upon the degree of error in their data (e.g. Fig. 4). Were instrument default parameters used in Ct calls, or did the authors adjust these parameters? Again, please state how this was set on the instrument in the Methods section. The cDNA reactions and Ct values illustrated in Fig 3 appear to be calling the Ct rather inconsistently between the samples, and this would affect the ratio and perhaps improve the error estimations. The authors should consider that Ct values over 32-33 are considered to be late amplification and may be non-specific, or at least represent an extremely low target cDNA copy number. The authors could confirm that the high Ct values are not artefactual with e.g. information from melt curves. The primers and reactions perform well using cDNA generated from pathogen RNA, but the amplifications in wood-extracted samples appears to be very weak.

Paragraph #5 of the Discussion L 345-376 should be re-written for clarity and consistency. The paragraph is very long, and wanders a bit from the first sentence (L345-346), a rather simplistic declaration, to later discussion of the reliance of the technique on the kinetics of mRNA stability (L 371 to 376). The paragraph could be reduced in length and the paragraph structure improved for clarity of the points made.

In addition, a few minor edits to the manuscript are recommended:

L63 insert “in eukaryotes”

L 134-136 “falcon” filters and tubes- Falcon is a brand not a type of tube- e.g. describe as “50 mL conical plastic centrifuge tube” or some such.

L 161 “Qubit fluorometer” list source

L 218 How was amplification efficiency calculated from the regression curves?

L236 synthesized (sp)

L 239 nucleic acids (no hyphen)

L255 ...efficiently killed... (reverse order of words)

Reviewer #5: The authors report new cDNA-based reverse transcription quantitative real time PCR (RT-qPCR) assays to detect the presence and viability of Phytophthora ramorum and Grosmannia clavigera. The authors test the ability of the assay to discriminate between living and dead fungal samples after two heat treatments. Additionally, the assay is used to detect the pathogens in artificially inoculated wood bolts.

The main merit of the publication is in developing molecular assays that can distinguish between dead and living mycelium based on cDNA/gDNA ratios. Assays that can evaluate pathogen viability have potential to improve phytosanitary measures. The P. ramorum assay is promising as a diagnostic tool, as it only amplifies P. ramorum DNA, and not other closely related species from Phytophthora clade 8c. However, the G. clavigera assay is not fully specific as it amplifies a closely related species: L. longiclavatum. This is problematic in terms of specific pathogen identification for diagnostic purposes. The impact of the work would be higher if the assay was specific for G. glavicera. Alternatively, the authors could develop an assay that robustly detects several closely related Leptographium/Grossmannia species.

The experimental design that was used to test the assay has some flaws and inconsistencies:

- The data for the heat treatments may not be fully comparable. Mycelial samples were processed differently for the two heat treatments: in kiln-drying, the authors used 0.1 ml tubes in a heat block, whereas in short heat treatment, they used agar plates in an oven. This might affect fungal viability and thus RNA degradation. The authors should provide data that shows this has no effect on the Cq values. Now, the ANOVA cannot distinguish if this affects the cDNA/gDNA ratios.

- No control mycelium samples for the different time points. As this is a time-point assay, the authors should include a non-treated control mycelium sample for each time point, and store them in identical conditions as the treated samples. This would tell if the expression of the gene target is actually stable in the used conditions, and would help to evaluate the suitability of this assay as a diagnostic tool.

- Wood samples were not plated to reisolate the pathogen. This would provide a concrete reference point to discuss the ability of the assay to detect the pathogens, either viable or dead.

- Wood samples were not compared for the heat treatments. This would have been an excellent simulation for the relevance of the assay as a diagnostic tool. Now the work falls short of demonstrating the applicability of this assay in phytosanitary screening.

- Imbalanced sample sizes for the two pathogens: 8 P. ramorum isolates, but only 1 G. clavigera isolate. In my opinion, the authors should have used more G. glavigera isolates.

Additionally, there are some concerns related to RNA sample quality control. The authors only used Nanodrop to quantify the RNA samples before DNAse treatment, which can distort the downstream RNA amounts added to the reverse transcription reactions. They used a kit that selects for RNA, but still, fluorescent RNA-binding dyes are the preferred method to get accurate RNA concentrations. Additionally, the authors do not report any method that was used to estimate RNA integrity (Nanodrop is not sufficient).

The manuscript has potential, but I cannot recommend the acceptance of this manuscript in its current form. The authors would need to make complementary experiments and provide more data to improve the manuscript. Especially I would like to see data for the effect of the heat treatment on the wood samples and pathogen viability, and how the assay performs as a diagnostic tool. Alternatively, the authors should re-analyze the data and re-write it as a short communication and tone down the role of the heat treatment comparison. This would in my opinion require excluding/re-developing the G. clavigera assay, as it is not species specific, but it is not a broad-spectrum assay either.

Comments and revisions

See PDF for language revisions

Please read Bustin et al. 2009 (10.1373/clinchem.2008.112797) for recommended terminology and experimental guidelines for quantitative PCR experiments

Correct real-time PCR to quantitative real-time PCR (qPCR)

Correct reverse transcription real-time PCR (RT-PCR) to reverse transcription quantitative PCR (RT-qPCR)

Correct Threshold cycle/Cycle threshold (Ct) to Quantification cycle (Cq) throughout the manuscript

Resolution of all figures is poor. Make sure you provide figure files with at least sufficient resolution (TIFF files with 300-600 DPI) and export graphs from R-studio/statistical software into EPS format to get 300 DPI resolution (see journal guidelines for preparing the images).

L47: What do you mean by universal genes? Conserved genes?

L78: How many conserved proteins, how many sequences?

L85-88: Rephrase to list P. ramorum as the primers were also tested on P. ramorum.

L94: Rephrase to list G. clavicera as the primers were also tested on G. clavicera.

L98-99: Provide citation to manuals/protocols/guidelines that were used to optimize the assays.

L113-131: The experimental setting is unclear. Please make an illustration showing the workflow, with heat treatments, what samples were used for what, differences in methods, replicates, controls, plating etc.

L103-104: Why did you decide to use 8 isolates of P. ramorum and only one G. clavicera isolate? This is a rather unbalanced experimental setting.

L103-112: How many bolts per each tree species-pathogen-time point combination?

L102-112: Why did you not plate any of the wood tissue? This would provide a reference point to evaluate what the re-isolation success is in controls vs. after heat treatment, and if the pathogens remain viable after the treatments. The manuscript would be improved if the authors provided results for re-isolation from inoculated wood samples.

L102-112: Why you did not use any of the wood samples for the heat treatments? This would be the closest to a real life simulation to evaluate the efficacy of the heat treatments. Now the results have very little connection to using the assay in a real-life diagnostic context.

L105: Provide ingredients or citation for carrot and MEA media

L118: Provide ingredients or citation for V8 and MEA media

L114-123: At what temperature were the samples stored after the SPF kiln-drying treatment before sample collection?

L120-121: Was the frozen mycelium sample in 1.5 ml tubes the only control? Did you have untreated controls stored in 0.1 ml strip tubes that were stored in same conditions as the heat treated samples, and sampled at the same time points? Flash freezing preserves mRNA better compared to a situation where a small non-heat-treated sample of mycelium is stored at room temperature. Now it is not clear how much RNA degradation occurs due to storage at room temperature, and how much is due to heat treatment. Additionally, knowing the expression of the gene over several time points would help to evaluate the suitability of this gene as a diagnostic marker. Please provide data for cDNA/gDNA ratios for control mycelial samples that are stored and sampled similarly to the heat-treated samples.

L127-129: For the short heat treatment, did you put agar plates directly to the oven? This might be a problem in terms of comparing the heat treatments. The mycelium on the plates might be heated differently than in the SPF kiln-drying treatment, where 0.1 ml Eppendorf tubes were used for heating and storing the samples after the treatment. Can you provide data that would indicate this won’t affect the degradation rate of mRNA? I would repeat this experiment by placing the mycelium into 0.1 ml Eppendorf tubes, perform the short heat treatment, and sample similarly as in the SPF kiln-drying experiment. With the current methods, I am not convinced that the data from the two heat treatments is comparable. The ANOVA won’t be able to tell if the differences in cDNA/gDNA ratios are caused by using 0.1 ml tubes vs. plates, or due to different temperature treatment, as the use of tubes or plates is nested within the heat treatment.

L159-163: Why did the authors decide to use two different methods for quantification of nucleic acids, Qubit for DNA and Nanodrop for RNA? Per MIQE guidelines, the preferred method for quantifying RNA uses fluorescent RNA-binding dyes, e.g. Qubit assay. As the RNA is not pure mRNA and the concentrations were measured before DNAse treatment, it is likely that Nanodrop readings overestimate the concentrations. This can affect the amount of RNA that is added to each RT reaction. Please address this issue and provide data that shows you get the similar Cq results from RNA samples measured with Qubit.

L159-163: What method did you use to evaluate RNA integrity? The absorbance values from Nanodrop only indicate RNA sample purity, not integrity.

L170-176: How many biological and technical replicates?

L204-208: Based on previous data, does the gene have stable constitutive expression? For diagnostic assays like the authors are developing, the gene targets should have stable expression no matter the condition. Add eg. text and references from lines 337-339.

L221-223: Figure 2 caption: Revise so that you provide the same information for both targets and indicate clearly which gene target is for which pathogen.

L234-238: Figure 3 caption: Indicate what the shaded areas are.

L250-261: Report results from statistical testing, as they are not indicated in Figure 4 or elsewhere.

L255-257: Less efficient compared to what: control, the other treatment? Be more specific.

L267-268: Table 1 indicates that replicate has a significant impact on G. clavicera cDNA/gDNA ratios. Discuss why this is observed for G. clavicera but not for P. ramorum? How was the quality of the RNA samples? How about lesion lengths, any variation in that?

L272-275: Summarize the results in 1-2 sentences in appropriate context. This is not re-isolation because what you describe is plating mycelium samples prepared from the same known pure culture. Rather you are assessing viability of mycelium after heat treatment.

L272-275 & L345-376: If I understand correctly, you did not make any re-isolations from the bolts. Discuss/Explain why not? Discuss if plating only the heat-treated mycelium samples provides a reliable reference for the viability of the pathogen in biological samples. How do the Cq values for mRNA in colonized bolts and mycelial samples compare? Based on the Cq values, what is the extent of mycelial colonization in the wood samples?

L312: At least in the US, by definition the pathogen is not regulated, but the interstate movement of certain articles from quarantined counties is regulated. See for details: https://www.federalregister.gov/documents/2007/02/27/07-892/phytophthora-ramorum-quarantine-and-regulations

L367-376: Be a bit critical and discuss if this could be resolved by plating infected wood samples after the heat treatments. There might be a time window after the treatment when the pathogen would still be able to grow, if suitable conditions (e.g. nutrients, temperature, moisture) were available. In what conditions are heat-treated/kiln-dried samples stored in industrial facilities? Is there a risk that the pathogen would still remain infectious? Does this create a risk for false-negatives? When should the diagnostic testing be done? Is this feasible?

Discussion and criticism lacks entirely for the non-specific G. clavigera assay. Discuss what diagnostic implications this might have, if you are not willing to expand the work and develop an assay that is specific for this pathogen. Are all Grosmannia/Leptographium species equally harmful? Could this assay be a more generic assay that detects several of these species?

 </comment></comment></comment></comment></comment>

6. PLOS authors have the option to publish the peer review history of their article (what does this mean?). If published, this will include your full peer review and any attached files.

Reviewer #1: No

Reviewer #2: No

Reviewer #3: No

Reviewer #4: No

Reviewer #5: No

---

## [Author Response · Author response to Decision Letter 0]

27 Nov 2019

RESPONSE TO REVIEWER’S COMMENTS (A Word copy is also attached, with color code to make it easier to follow the comments and answers).

Reviewer #1: 

The experiments are adequate and compelling for most of the work, but I noticed a major discrepancy between what is claimed to be done and what is reported in the mat&met and results sections. In this respect, I recommend a major revision.

Indeed, the abstract states that “a stability analysis was conducted by comparing the ratio of mRNA to gDNA overtime following heat treatment of wood infected by the oomycete..”. 

Corrected in manuscript. “wood infected with” was deleted L.26.

Also P12L244, it is written: “Ratio of ....applied to WOOD-LOGS infected with P. ramorum and G. clavigera”. However, in the experiments reported in the manuscript, only mRNA and gDNA from MYCELIAL cultures of the pathogen were heat-treated, not from WOOD infected by the latter (see §Heat treatment to determine mRNA stability”, 

P6-7; in Figure 4 caption “... for gDNA or cDNA extracted from CULTURES sampled from 0 to 240 hours after treatment.”), and in discussion section L285-287, L382.

Likewise, all the discussion parts related to the appropriateness of wood treatments deserve mitigation since the experiments carried out in this work only dealt with in vitro-cultivated mycelium and not in vivo growing mycelium. The efficiency of heat treatment procedures are dependent of wood thickness, relative humidity, heat flow, etc.

I therefore suggest clarifying what was really assessed in this work. It seems that artificially infected wood were indeed tested after 28 days of incubation, but not after the two different heat treatments.

We clarified what was done in our work i.e. all statements talking about heat treatment on infected wood have been removed and the discussion section on this point has been toned down. See modifications on L.26, L.284, L.367...

Specific comments:

L46: Ref 8 and 9 do not seem to be the most relevant ones to support the statement of the sentence.

We changed these two references for : Taylor et al. 2000 Phylogenetic Species Recognition and Species Concepts in Fungi. Fungal Genetics and Biology, 31:21-32

L50: again here, ref 11 does not seem to be appropriate to the support the statement.

Right. We removed ref. 11 from L.56.

L69: ref 21 and 22 are more appropriate here, ref 20 a bit less supporting. Consider removing it.

Done. We removed ref. 20.

L79 and 90: please could you explain why distinct DB were used for the selection of genes?

As Grosmannia is a fungus, it was searched against a fungal database i.e. FUNYBASE. Unfortunately, searching against this db couldn’t work for Phytophthora as these organisms are not fungi. For this reason, we searched Phytophthora against a more generalistic db i.e. the CEGMA eukaryotic db. 

L82: on which basis the genes were selected? It is written “genes are expected to be expressed”, but how was that anticipated? The discussion section contains important information pertaining to the rationale followed for the selection, and this information should be moved to the mat&met section.

Done. See L.90-91 in M&M section.

L133: number, reference of isolates and number of replicates should better be indicated here, rather than in a later paragraph (L177-179).

This paragraph was moved up to the beginning of the M&M section entitled “Chlamydospore separation” (L.146-148).

L140-141: the sentence sounds a bit odd.

We changed the sentence for “Final concentration of the spores was adjusted to 1x105 chlamydospore/ml using a hemocytometer”. L.156-157.

L147: please specify what the wood samples were collected from.

Done. See L.163-164.

L177: how was the qPCR done with these solutions as templates? 2.2 ng of template DNA are indicated a few lines above. Is it still the case?

Yes, for all TaqMan reaction DNA templates were adjusted to 2.2ng (L.191). 

Figure 1 caption indicates that gDNA and cDNA were used, but it is in conflict with what is written L215-216.

L215-216 (L.237 in the new version with track changes) was referring to Figure 2. We are not sure about what Rev. #1 is referring to with this point.

L201: is there one single sister species (L. longiclavatum) or several involved here? Not clear what “these sister species” refer to.

No, four of them but the closest one is L. longiclavatum. We change “its sister species” for “the closely related species” to discard any ambiguity (L.217).

Figure 3 should remind how many replicates were included, and also explain how the curves should be interpreted (shaded areas, min and max?). Also, it should be interesting to discuss the striking lower magnitude of fluorescence yielded with cDNA extracted from P. ramorum-infected Tsuga, whereas the gDNA curves look “normal”. I noticed that there was a change of y-axis scale compared to the other plots, probably in order to better fit the gDNA fluorescence magnitude. But still, the gDNA/cDNA difference with the other plots seems awkward.

We re-wrote Fig. 3 (NOW FIG. 4) caption as follows (L.645): Figure 4. Wood inoculation with Phytophthora ramorum (EU2 isolate P2111) and Grosmannia clavigera (isolate KW140) and corresponding real-time PCR amplification plots. For each wood inoculation sample, gDNA and cDNA synthetized from mRNA extracted from a lesion at 28 day post inoculation was tested in real-time PCR with either the PH178 assay (targeting P. ramorum) or MS356 (G. clavigera). Quantification cycle (Cq) values for gDNA (blue) and cDNA (red) are reported on each graph. Shaded area around the average represents ±SD.

This difference for gDNA vs. cDNA with Tsuga is probably the result of an inhibition of the cDNA synthesis. Unfortunately, we did not keep a record of the RNA quality after purification from wood samples and cannot verify this hypothesis. 

L239: I do not think that it is relevant to compare the mean Ct values obtained with chlamydospores to those obtained with infected wood samples. Chlamydospores quantity was arbitrarily set, we do not know if is close to reality in infected tissue or soil for instance.

Corrected in manuscript, comparison sentence removed. L.251-252.

Table 1: consider removing it, as it is a piece of arid stats. Or maybe replace “treatment” with what it stands for.

Table 1 has been removed.

L285: heat treatment may just not degrade mRNA, it may also inhibit its synthesis. 

We considered this important point and change the original sentence for “As a second method, we assumed that the rapid degradation and synthesis inhibition of mRNA might be a useful indicator of cell mortality [47]”. L.387.

L367-L372: I fully agree with this caveat. The authors should therefore recommend a practical procedure to assess the kinetics of mRNA.

There are two ways of addressing this issue: one if to determine the threshold after which mRNA is not detectable after lethal treatments so that testing can be done at the appropriate time. The second way would be to conduct for each targeted species an mRNA kinetics experiment to determine the rate of degradation. However, this last solution could be complicated by the fact that mRNA kinetics could be affected by the substrate treated. 

￼Figure 4: The authors should discuss some questioning variations observed in cDNA quantity, e.g. the high variation in cDNA amplification between Ctrl/ 0 and 24 hrs post treatment for P. ramorum/short heat treatment. Although it seems obvious that after a certain treatment duration, no mRNA is produced, what would explain this high variation before that time? One may also wonder why gDNA and cDNA quantities (inferred from Ct values) in the control are sometimes significantly superior to the 0 tpt values? In two occasions, no mRNA is detected at all for 0 tpt, whereas the ctrl value is “normal”. Is it because 0 means that the treatment already started? If yes, 0 is probably not appropriate term.

The time points refer to the time after treatment, therefore, the “0” time point actually refers to when the pathogen has just been removed from their corresponding heat treatments. The x-axis for this figure was labeled to indicate that the time points 0 to 240 are the number of hours after treatment. 

High variation in Ct values could be due to the difference in yield of mRNA obtained after purification (see Discussion on L.402-408) and the instability of RNA (i.e. RNA degradation).

More general comments:

The discussion section is a lot about P. ramorum, and unfortunately G. clavigera is a bit set aside. Although G. clavigera is less “famous” than P. ramorum, a more balanced discussion would be nice.

We tried to add a few more statements on G. clavigera. However, this pathogen didn’t receive the focus that P. ramorum got in terms of tool development for monitoring and surveillance, making it very difficult to comment/discuss on G. clavigera.

Reviewer #2: 

The discussion could have more information comparing other methods and talk about the limitation of the assays, by example the detection of mRNA is very at the Ct value of limit of detection. What about other assays published. Other missing or clarifying information need to be add.

Yes, further discussion of the limitations to this assay was added (Paragraph starting L.398). The detection of some of the samples were at a very low Ct, but still within our cut off of 35, eliminating the possibility of false positives. 

Specific comments:

P4L58 did you consider to add other method information to look at viability, I believe the PMA Propodium monoazide is a good method and do not need to play with RNA. Should add info and pro and cons in discussion?

Additional information has been added to the manuscript to address this point. Paragraph starting on L.398.

P4L69 I was wondering about if the P. ramorum was infecting in wood or surface but was more clear later in manuscript. Could be presented more here. P5L82, it is really a communication or more a resource? (Bret Tyler sequences)

Sentence has been changed to : “...two microorganisms that can colonize wood cambium”. L.75-76.

P6L104 for G. clavigera, only one isolate was use, how do we know it would work with multiple genotypes?

Yes, this is one limitation of our study. The isolate was chosen because this was the isolate that was used for genome sequencing. However, G. clavigera is highly clonal, meaning that sequence polymorphism is limited in conserved genes and we made sure that the primers developed were matching on other G. clavigera sequenced.

P6L109 any reason on use of only conifers, what about deciduous tree for P. ramorum?

Canadian forest sector mainly focus on conifers and the industry predominantly deals with the trading of such (e.g. coniferous forests account for 72% of timber volume in Canada). 

P6L115 any specific reference for the SPF kiln drying schedule?

We added the following reference on L. 129: 

Cai, Liping, and Luiz C. Oliveira. Evaluating the use of humidification systems during heat treatment of MPB lumber. Drying Technology 29, no. 7 (2011): 729-734.

P8L144 "Matric" should be "Matrix" Correct in line 154.

Corrected in manuscript. L.160 and L.160.

P9L178 "586 and 101329) info on specimens, it is voucher isolate, CBS culture??? need more info on you isolates used in the manuscript?

Pr-05-015 (CSL2268, P1578) was isolated from Rhododendron grandiflora from a nursery in the UK in 2002 and is Clonal lineage EU1.

CBS101329 was purchased from CBS-KNAW Fungal Biodiversity Centre (now known as Westerdijk Fungal Biodiversity Institute). It was collected from Lisse, Netherlands on Rhododendron and is of the EU1 strain. We added this information on L.147.

P9 Just wondering it is better ration Ct or Ration concentration calculated, Ct,Log10?

We are not sure we understand this question. Does Rev. #2 suggest to log-transform Ct-ratios? 

However, we think it’s better to use the ratio of the Ct-values obtain for cDNA and gDNA (in theory, cDNA Ct-value should increase with loss of viability whereas gDNA Ct-value shouldn’t change).

P12L232, 33.4 seem to be very low limit? not so high you may miss lower concentration of pathogens. also should explain the range in figure 3, blurry red??

￼Blurry blue, same Ct but range blurry?? not clear what it mean?

Redefinition of figure 3 was done (NOW FIG. 4) - please refer to comment from Rev. #1.

P12L247 and Table 1, not so clear what the F, Fvalue is the stat need to be better explained and significance in the manuscript?

Table 1 no footnotes, what the "*" means? By? Type III SS?

Yes, Table 1 is deleted as suggested by Rev. #1.

Discussion, RNA extraction and cDNA is additional step, add info how the RNA extraction is easy or difficult and cDNA prep, risk of contamination, compare to PMA should be discuss.

Additional information has been added to the discussion that the RT-qPCR conducted here was a two-step process. Paragraph starting on L.398.

What is the Ct level limit for cDNA-mRNA in other publication using similar approach?

Similar papers (e.g. Chimento et al 2012, Leal et al 2013) used a Ct cut-off value later than 35.0 as their positive cut off. 

Reviewer #3: 

Effectiveness of the heat treatments on the pathogens growing in the wood trunk was, however, not evaluated. The author needs to address this point. Below are my specific comments.

Same point than Rev. #1. As stated above, we clarified that the effectiveness of the treatment was tested only on cultures.

Line 18: RNA represents ... therefore only be produced by living organisms. RNA, as well as DNA, are only produced by living organisms.

Right. We fixed that by changing the sentence: “RNA represents the transcription of genes and can therefore become rapidly unstable after cell death…”. L.20-21.

Line 164: cDNA synthesis

The authors used two-step RT-qPCR for all samples. The two-step method, which uses a stock cDNA, may be advantageous for the screening of primer pairs. However, for high-throughput assay, one-step RT-qPCR is more advantageous and give less experimental variation.

A new section has been included in the discussion to address this point. But overall, using the one-step method is less sensitive and requires to start with more RNA and though could be considered for future improvement. Paragraph starting on L.398. 

Line 231: Cycle-threshold values obtained for the cDNA were slightly higher.

For both P. ramorum and G. clavigera, the differences of Ct values between gDNA and cDNA were over 3, which were equivalent to an 8-fold difference in copy number of templates. This is a huge difference and needs to be discussed.

We added some elements of discussion (+ references) about this point in a new paragraph starting on L.398. Our main hypothesis is related to the low purity/yield of the RNA extracted from conifer wood and subsequent inhibition of downstream enzymatic reaction such as reverse transcription during cDNA synthesis.

Line 242: No amplification was observed in the no-template control.

The vast quantity of genomic DNA and cDNA from inoculated wood samples were of the host plant origin. What was the level of background amplification when a DNA template from a non-inoculated plant tissue was used?

Yes, gDNA and cDNA samples from non-inoculated wood were included in the qPCR but no expression was observed. This point is now included in manuscript L.253-254.

Line 342: indicating that the target gene is still expressed during this life-stage. Most likely, the target gene is not expressed in the resting spores, but its transcript is stored in them.

I’m not sure we could have some evidence for transcript storage in resting spores. Statement has been removed.

Reviewer #4: 

The real time PCR data depicted in Fig 3 is however a bit troubling, with regard to the Ct values. The authors should mention the instrument used in the assays in the Methods section, L 170-176, 

Done. Applied Biosystems StepOne™ software was used for the qPCR. L.194.

and how the thresholds were established for Ct calculations. 

The qPCR thresholds were generated through the StepOne™ default parameters. The parameters calculate the threshold, under “automatic Ct”. This is an analysis setting in which the software calculates the baseline start and end values and the threshold in the amplification plot. The software uses the baseline and threshold to calculate the threshold cycle (CT). A sentence was included in manuscript for clarification. L.194-195.

A more precise estimation of the Ct could improve upon the degree of error in their data (e.g. Fig. 4). Were instrument default parameters used in Ct calls, or did the authors adjust these parameters? 

Default parameters of the StepOne™ software were used for the Ct threshold calculations.

Again, please state how this was set on the instrument in the Methods section. The cDNA reactions and Ct values illustrated in Fig 3 appear to be calling the Ct rather inconsistently between the samples, and this would affect the ratio and perhaps improve the error estimations. The authors should consider that Ct values over 32-33 are considered to be late amplification and may be non-specific, or at least represent an extremely low target cDNA copy number. 

Details of the qPCR instrument has been included in the manuscript. L.194-195.

Though a Ct value of 32-33 is considered low as compared to gDNA samples. It is still within other similar studies’ positive amplification cut off (Chimento et al 2012, Leal et al 2013).

The authors could confirm that the high Ct values are not artifactual with e.g. information from melt curves. The primers and reactions perform well using cDNA generated from pathogen RNA, but the amplifications in wood-extracted samples appears to be very weak.

The optimization of wood extraction should help in improving the sensitivity of the assays. Due to the high amount of high quality of secondary metabolites polyphenols, tannins, terpenoids and protein inhibitors (see L.402-408). 

Unfortunately, the experiments were run under the standard curve settings on the StepOne qPCR machine (not under the melting curve settings), meaning that the size of the PCR fragments cannot be confirmed with the melting curves.

Paragraph #5 of the Discussion L 345-376 should be re-written for clarity and consistency. The paragraph is very long, and wanders a bit from the first sentence (L345-346), a rather simplistic declaration, to later discussion of the reliance of the technique on the kinetics of mRNA stability (L 371 to 376). The paragraph could be reduced in length and the paragraph structure improved for clarity of the points made.

The two first sentences of paragraph #5 were removed. L.363-366.

In addition, a few minor edits to the manuscript are recommended:

L63 insert “in eukaryotes” 

Done. 

L 134-136 “falcon” filters and tubes- Falcon is a brand not a type of tube- e.g. describe as “50 mL conical plastic centrifuge tube” or some such.

Corrected in manuscript. L.151.

L 161 “Qubit fluorometer” list source

L 218 How was amplification efficiency calculated from the regression curves? L236 synthesized (sp)

Efficiency was an output from the StepOne qPCR machine.

L 239 nucleic acids (no hyphen)

Corrected in manuscript. L.250

L255 ...efficiently killed... (reverse order of words)

Corrected in manuscript. L.267

Reviewer #5: 

The authors report new cDNA-based reverse transcription quantitative real time PCR (RT-qPCR) assays to detect the presence and viability of Phytophthora ramorum and Grosmannia clavigera. The authors test the ability of the assay to discriminate between living and dead fungal samples after two heat treatments. Additionally, the assay is used to detect the pathogens in artificially inoculated wood bolts.

The main merit of the publication is in developing molecular assays that can distinguish between dead and living mycelium based on cDNA/gDNA ratios. Assays that can evaluate pathogen viability have potential to improve phytosanitary measures. The P. ramorum assay is promising as a diagnostic tool, as it only amplifies P. ramorum DNA, and not other closely related species from Phytophthora clade 8c. However, the G. clavigera assay is not fully specific as it amplifies a closely related species: L. longiclavatum. This is problematic in terms of specific pathogen identification for diagnostic purposes. The impact of the work would be higher if the assay was specific for G. glavicera. Alternatively, the authors could develop an assay that robustly detects several closely related Leptographium/Grossmannia species.

Leptographium longiclavatum and G. clavigera are closely related species (i.e. two SNPs over a 710nt alignment of the ITS). They share similar morphological characteristics and evolutionary history (see Lee et al. 2005 Mycol. Res. 109:1162; Lims et al. 2004 FEMS Microbiology Letters, 237, 89). This made it very complicated to find a good tradeoff between finding a conserved enough gene in the FUNYBASE db. to anchor primers arround a intron-exon junction and enough fixed polymorphisms to be able to differentiate between L. longiclavatum and G. clavigera. As stated in the ms on L.211-214 our assay now targets both species. We added this to the Discussion (L.330-342): ‘Due to the high sequence similarity between G. clavigera and the closely related species L. longiclavatum [62,63], the assay developed amplified mRNA and gDNA of both species. This could be a positive attribute of this assay since both species are symbionts of the mountain pine beetle and share a very similar ecological niche in pine trees [22]; in fact it is possible that these fungi hybridize and the species limits are not clear (R. C. Hamelin and A. Capron, unpublished).’ 

The experimental design that was used to test the assay has some flaws and inconsistencies:

- The data for the heat treatments may not be fully comparable. Mycelial samples were processed differently for the two heat treatments: in kiln-drying, the authors used 0.1 ml tubes in a heat block, whereas in short heat treatment, they used agar plates in an oven. This might affect fungal viability and thus RNA degradation. The authors should provide data that shows this has no effect on the Cq values. Now, the ANOVA cannot distinguish if this affects the cDNA/gDNA ratios.

Yes, this is true. However, in both cases, the targeted temperature for the core is reached in both experiments; as long as this temperature is reached, we think that we can assume that the temperature requirements of the experiment were fulfilled. 

In addition, the point of the experiment was not to compare both experiments. Instead, the aim was to use our assays to assess if some temperature/schedule used to treat wood against pests are efficient. 

- No control mycelium samples for the different time points. As this is a time-point assay, the authors should include a non-treated control mycelium sample for each time point, and store them in identical conditions as the treated samples. This would tell if the expression of the gene target is actually stable in the used conditions, and would help to evaluate the suitability of this assay as a diagnostic tool.

Right, there was no non-treated control for each of the different sampling time points after heat treatment and we didn’t assess the stability of the expression of the targeted genes in non-treated conditions. However, there was a non-treated control mycelium that was frozen at the end of experiment and tested with the gDNA and cDNA assays. For both species it resulted in detectable Ct-values for the cDNA meaning that the targeted gene was expressed during the experiment if the culture was not treated. 

- Wood samples were not plated to reisolate the pathogen. This would provide a concrete reference point to discuss the ability of the assay to detect the pathogens, either viable or dead.

Right, they were not plated for this experiment. However, the aim of this experiment was very “technical”. We wanted to test if gDNA and mRNA of P. ramorum and G. clavigera could be 1 - purified from inoculated wood from different conifers and 2 - detected by qPCR with our assays. 

- Wood samples were not compared for the heat treatments. This would have been an excellent simulation for the relevance of the assay as a diagnostic tool. Now the work falls short of demonstrating the applicability of this assay in phytosanitary screening.

Similarly to what was discussed above in response to Rev. #1, we clarified what was really done in our work i.e. all statements talking about heat treatment on infected wood have been removed and the discussion section on this point has been toned down. Though heat treated wood samples were not tested on using the developed assays, the ability of the assays on infected woody shows promise in field trials.

- Imbalanced sample sizes for the two pathogens: 8 P. ramorum isolates, but only 1 G. clavigera isolate. In my opinion, the authors should have used more G. glavigera isolates.

Already answered above. See Rev. #2

Additionally, there are some concerns related to RNA sample quality control. The authors only used Nanodrop to quantify the RNA samples before DNAse treatment, which can distort the downstream RNA amounts added to the reverse transcription reactions. They used a kit that selects for RNA, but still, fluorescent RNA-binding dyes are the preferred method to get accurate RNA concentrations. Additionally, the authors do not report any method that was used to estimate RNA integrity (Nanodrop is not sufficient).

An additional sentence was included in manuscript regarding RNA integrity (L.180-181) and discussion on the use of RNA binding dyes as a consideration in future testing (L.416-420).

The authors would need to make complementary experiments and provide more data to improve the manuscript. Especially I would like to see data for the effect of the heat treatment on the wood samples and pathogen viability, and how the assay performs as a diagnostic tool. Alternatively, the authors should re-analyze the data and re-write it as a short communication and tone down the role of the heat treatment comparison. This would in my opinion require excluding/re-developing the G. clavigera assay, as it is not species specific, but it is not a broad-spectrum assay either.

As explained above this study is a proof of concept to demonstrate that we can target and differentiate by qPCR gDNA and mRNA (cDNA) in two wood-infecting pathogens. At the same time, we used these assays to look at the efficacy of heat-treatment in killing these pathogens. We agree that some additional experiments would be required to assess the efficacy of the wood treatment on infected wood sample. Unfortunately, we didn’t have the facility to conduct such an experiment on wood logs and/or lumber with a regulated pathogen like P. ramorum. 

Please read Bustin et al. 2009 (10.1373/clinchem.2008.112797) for recommended terminology and experimental guidelines for quantitative PCR experiments Correct real-time PCR to quantitative real-time PCR (qPCR)

Done

Correct reverse transcription real-time PCR (RT-PCR) to reverse transcription quantitative PCR (RT-qPCR)

Done

Correct Threshold cycle/Cycle threshold (Ct) to Quantification cycle (Cq) throughout the manuscript

Done

Resolution of all figures is poor. Make sure you provide figure files with at least sufficient resolution (TIFF files with 300-600 DPI) and export graphs from R- studio/statistical software into EPS format to get 300 DPI resolution (see journal guidelines for preparing the images).

All our figures were 600dpi. The conversion of our files in pdf format for review purpose might have reduced the resolution of the figures? 

L47: What do you mean by universal genes? Conserved genes? 

We meant “conserved” genes that are found in all eukaryotic species. We change it for “conserved”. L. 52.

L78: How many conserved proteins, how many sequences?

228. See L.86.

L85-88: Rephrase to list P. ramorum as the primers were also tested on P. ramorum.

Done. L.97-98.

L94: Rephrase to list G. clavicera as the primers were also tested on G. clavicera.

Done. L.104.

L98-99: Provide citation to manuals/protocols/guidelines that were used to optimize the assays.

There was a confusion with the sentence “All assays were then optimized for use in identifying mRNA stability.” We apologize for this mistake.

Due to the constraint of having the qPCR probe located on an intron-exon junction, the assays were not optimized after their design and first testing. The only optimization done was for the primers, by following the recommendations made in Feau et al. 2018 PeerJ 6:e4392.

L113-131: The experimental setting is unclear. Please make an illustration showing the workflow, with heat treatments, what samples were used for what, differences in methods, replicates, controls, plating etc.

Let me know what you think and suggestion for improvement! 

Figure caption: Work flow diagram of the short (left) and long kiln (right) heat treatments. Orange plates represents P. ramorum on V8 agar, while yellow plates as G. clavigera on malt extract agar (MEA).

Below is the draft workflow - this figure was added to the ms as a new Figure (Fig. 2; L. 129):

L103-104: Why did you decide to use 8 isolates of P. ramorum and only one G. clavicera isolate? This is a rather unbalanced experimental setting.

Our study is a proof-of-concept that the inoculated pathogen could be detected in wood samples. To do so, we only needed one individual of G. clavigera (and we picked the one that was used for genome sequencing). In addition, G. clavigera is highly clonal, and therefore our assay should also work on other genotypes.

L103-112: How many bolts per each tree species-pathogen-time point combination?

Two bolts were used for each tree species-pathogen-time point combination. 

L102-112: Why did you not plate any of the wood tissue? This would provide a reference point to evaluate what the re-isolation success is in controls vs. after heat treatment, and if the pathogens remain viable after the treatments. The manuscript would be improved if the authors provided results for re-isolation from inoculated wood samples.

Same point than the fourth comment made by Rev. #5 above. Right, they were not plated for this experiment. However, the objective of the wood-inoculation exp. was to test if gDNA and mRNA of P. ramorum and G. clavigera could be 1 - purified from inoculated wood from different conifers and 2 - detected by qPCR with our assays. There was no need to re-isolate the pathogens from wood for validating that.

L102-112: Why you did not use any of the wood samples for the heat treatments? This would be the closest to a real life simulation to evaluate the efficacy of the heat treatments. Now the results have very little connection to using the assay in a real-life diagnostic context.

Yes, this is true. However, we did not have the facility & material to be able to reproduce the two heat treatment protocols with inoculated wood logs. As this is a proof-of-concept study we limited it to 1 - testing if we were able to amplify by PCR gDNA and mRNA (cDNA) extracted from inoculated wood and 2 - using our assay to determine if heat treatment protocols are accurate enough to kill mycelial cultures of G. clavigera and P. ramorum. Testing the same protocols on inoculated material will be the next step once we will be able to manage heat treatment protocols on infected wood products (logs or lumber).

L105: Provide ingredients or citation for carrot and MEA media L118: Provide ingredients or citation for V8 and MEA media

Done. L.116.

L114-123: At what temperature were the samples stored after the SPF kiln-drying treatment before sample collection?

Samples were left at room temperature and a sentence has been added to clarify this point in the manuscript. L.135.

L120-121: Was the frozen mycelium sample in 1.5 ml tubes the only control? Did you have untreated controls stored in 0.1 ml strip tubes that were stored in same conditions as the heat treated samples, and sampled at the same time points? 

Flash freezing preserves mRNA better compared to a situation where a small non- heat-treated sample of mycelium is stored at room temperature. Now it is not clear how much RNA degradation occurs due to storage at room temperature, and how much is due to heat treatment. 

Additionally, knowing the expression of the gene over several time points would help to evaluate the suitability of this gene as a diagnostic marker. Please provide data for cDNA/gDNA ratios for control mycelial samples that are stored and sampled similarly to the heat-treated samples.

As stated above, the frozen mycelium was the only untreated control sample for this experiment. For both species it resulted in detectable Ct-values for the cDNA meaning that the targeted gene was expressed during the experiment if the culture was not treated.

L127-129: For the short heat treatment, did you put agar plates directly to the oven? This might be a problem in terms of comparing the heat treatments. The mycelium on the plates might be heated differently than in the SPF kiln-drying treatment, where 0.1 ml Eppendorf tubes were used for heating and storing the samples after the treatment. Can you provide data that would indicate this wonʼt affect the degradation rate of mRNA? 

I would repeat this experiment by placing the mycelium into 0.1 ml Eppendorf tubes, perform the short heat treatment, and sample similarly as in the SPF kiln-drying experiment. With the current methods, I am not convinced that the data from the two heat treatments is comparable. The ANOVA wonʼt be able to tell if the differences in cDNA/gDNA ratios are caused by using 0.1 ml tubes vs. plates, or due to different temperature treatment, as the use of tubes or plates is nested within the heat treatment.

Answered above. The aim of the study was not to compare the two treatments. However, when the dataset was separated by treatments and compared with the untreated control (thus eliminating the affect or tubes vs plates), there is still a significant influence of treatment. Though it may not be statistically comparable, we can see that both heat treatments showed promise in eliminating the target pathogens.

L159-163: Why did the authors decide to use two different methods for quantification of nucleic acids, Qubit for DNA and Nanodrop for RNA? Per MIQE guidelines, the preferred method for quantifying RNA uses fluorescent RNA- binding dyes, e.g. Qubit assay. As the RNA is not pure mRNA and the concentrations were measured before DNAse treatment, it is likely that Nanodrop readings overestimate the concentrations. This can affect the amount of RNA that is added to each RT reaction. Please address this issue and provide data that shows you get the similar Cq results from RNA samples measured with Qubit.

DNA and RNA were extracted in two different labs (UBC for DNA and Canadian forest service (Victoria, BC) for RNA). At the time the RNA was extracted the CFS lab only had a NanoDrop instrument. We still do not believe that this affected the outcome of our experiments.

L159-163: What method did you use to evaluate RNA integrity? The absorbance values from Nanodrop only indicate RNA sample purity, not integrity.

Extracted RNA samples’ integrity was assessed using an agarose gel for visualization of the band fluorescence. An additional sentence was included in the manuscript. L.180-181.

L170-176: How many biological and technical replicates?

Additional sentence included for clarity. L176-177.

L204-208: Based on previous data, does the gene have stable constitutive expression? For diagnostic assays like the authors are developing, the gene targets should have stable expression no matter the condition. Add eg. text and references from lines 337-339.

Additional sentence included for clarity. L.225-227.

L221-223: Figure 2 caption: Revise so that you provide the same information for both targets and indicate clearly which gene target is for which pathogen.

Corrected. Figure 2 is now Figure 3. Its caption reads as follows:

“ Log-transformed standard curve assessed with gDNA serial serial dilution (1:10) of (A) Phytophthora ramorum for the TaqMan probe PH178_EX (A) (R2= 0.992 Eff%= 99.33) and (B) Grosmannia clavigera for the TaqMan probe MS359_EX (B) for the TaqMan probe PH178_EX and MS359_EX. (R2= 0.981 Eff%= 100.303). ”

L234-238: Figure 3 caption: Indicate what the shaded areas are.

Figure caption rewrote. See answer to Rev. #1

L250-261: Report results from statistical testing, as they are not indicated in Figure 4 or elsewhere.

Tukey HSD test were done to compare means in each experiment. The results of these tests have been added to fig. 4 (Now, Fig. 5). 

L255-257: Less efficient compared to what: control, the other treatment? Be more specific.

Corrected in manuscript.

L267-268: Table 1 indicates that replicate has a significant impact on G. clavicera cDNA/gDNA ratios. Discuss why this is observed for G. clavicera but not for P. ramorum? How was the quality of the RNA samples? How about lesion lengths, any variation in that?

Table 1 has been removed as suggested by Rev. #1.

L272-275: Summarize the results in 1-2 sentences in appropriate context. This is not re-isolation because what you describe is plating mycelium samples prepared from the same known pure culture. Rather you are assessing viability of mycelium after heat treatment.

Corrected and section heading redefined.

L272-275 & L345-376: If I understand correctly, you did not make any re- isolations from the bolts. Discuss/Explain why not? Discuss if plating only the heat-treated mycelium samples provides a reliable reference for the viability of the pathogen in biological samples. 

Yes, the attempt to re-isolate from woody samples will provide more insight on the difficulties in “culturability” from environmental samples. That being said, the re-isolation from the bolts was not a part of the scope of the study, as the bolts were not heat treated.

How do the Cq values for mRNA in colonized bolts and mycelial samples compare? 

Based on the Cq values, what is the extent of mycelial colonization in the wood samples?

It is difficult to compare the mRNA Cq values of the wood and mycelial samples as they were not of the same concentration and it is more difficult to extract woody tissue. Optimization of the woody extrataction, followed by dilution to the same concentration would be required before we can make the comparison of the Cq values. However, we can expect there to be a lower level of detection as it is usually difficult within environmental samples, especially that of wood with high levels of secondary metabolic polyphenols tannins, terpenoids and protein inhibitors. See the new paragraph about “limitations” of our study starting L. 398.

L312: At least in the US, by definition the pathogen is not regulated, but the interstate movement of certain articles from quarantined counties is regulated. See for details: https://www.federalregister.gov/documents/2007/02/27/07-892/ phytophthora-ramorum-quarantine-and-regulations

Thank you for this information. We clarified our statements accordingly on L.315-319.

L367-376: Be a bit critical and discuss if this could be resolved by plating infected wood samples after the heat treatments. There might be a time window after the treatment when the pathogen would still be able to grow, if suitable conditions (e.g. nutrients, temperature, moisture) were available. In what conditions are heat- treated/kiln-dried samples stored in industrial facilities? Is there a risk that the pathogen would still remain infectious? Does this create a risk for false-negatives? 

The experiment Rev. 5 is talking about here was about testing two heat treatments on mycelial cultures of G. clavigera and P. ramorum, not on infected wood. Mycelium samples were plated right after each heat treatment (at T = 0 post treatment) and no growth was observed for each of the cultures (L.277-278). This suggests that there’s no time window after the treatment when the pathogen is still able to grow, even if his cells could still be partly functional (this point was specifically Discussed on L.382-388). 

Even if these pathogens were able to grow after these treatments, industrial samples after heat/kiln-dried treatment are usually wrapped with plastic branding then kept outdoors. As such, cooler temperatures are probably not optimal for both pathogens (P. ramorum minimal growth temperature is 9°C with optima between 15 and 21°C; optimum growth temperature for G. clavigera is around 20°C). The mycelia samples in this experiment was kept at room temperature, which was more optimal for growth and no growth was obtained. 

When should the diagnostic testing be done? Is this feasible?

Ideally, the diagnostic testing should be done immediately after treatment, as this will be the best to eliminate the potential of false positives. From our experiment, we saw that the kiln treated samples showed no amplification of RNA, suggesting that this temperature coupled with the long exposure rapidly degrades the RNA. Therefore, if we detect amplification of RNA in these samples, it could be the presence of viable pathogens.

Discussion and criticism lacks entirely for the non-specific G. clavigera assay. Discuss what diagnostic implications this might have, if you are not willing to expand the work and develop an assay that is specific for this pathogen. Are all Grosmannia/Leptographium species equally harmful? Could this assay be a more generic assay that detects several of these species?

Grosmannia clavigera and Leptographium longiclavatum are both equally “harmful” as they are fungal symbionts to Mountain pine beetle and do not cause structural damage to the infected woody tissue but instead just a discolouration. We added a paragraph about G. clavigera in the Discussion section L.330-342.

---

## [Editor Report · Decision Letter 1]

3 Dec 2019

Molecular assays to detect the presence and viability of Phytophthora ramorum and Grosmannia clavigera

PONE-D-19-22766R1

Dear Dr. Richard Hamelin,

We are pleased to inform you that your manuscript has been judged scientifically suitable for publication and will be formally accepted for publication once it complies with all outstanding technical requirements.

With kind regards,

Simon Francis Shamoun, Ph.D.

Academic Editor

PLOS ONE

Additional Editor Comments (optional):

Dr. Richard Hamelin

Professor of Forest Pathology

University of British Columbia

Department of Forest Sciences

Vancouver, BC, Canada

Dear Dr. Hamelin,

I have reviewed your revised version of the manuscript entitled: "Number PONE-D-19-22766R1- "Molecular assays to detect the presence and viability of Phytophthora ramorum and Grosmannia clavigera" and found it acceptable for publication in PLOS ONE. Based on my expertise and the reviews of the five external reviewers, I recommend publishing your revised version of your manuscript in PLOS ONE.

At this time, on behalf of PLOS ONE, I would like to congratulate you and your co-authors on this outstanding research work and results. I commend you and your research team for this achievement.

We look forward to receiving future research articles from you and your research team.

Best regards,

Dr. Simon Francis Shamoun

Academic Editor

PLOS ONE
---

## [Editor Report · Acceptance letter]

15 Jan 2020

PONE-D-19-22766R1 

Molecular assays to detect the presence and viability of *Phytophthora ramorum* and *Grosmannia clavigera*

Dear Dr. Hamelin:

I am pleased to inform you that your manuscript has been deemed suitable for publication in PLOS ONE. Congratulations! Your manuscript is now with our production department. 

With kind regards,

on behalf of

Dr. Simon Francis Shamoun 

Academic Editor

PLOS ONE